# Smoothing-Based Conformal Prediction for Balancing Efficiency and Interpretability

## Abstract

Conformal Prediction (CP) is a distribution-free framework for constructing statistically rigorous prediction sets. While popular variants such as CD-split improve CP's efficiency, they often yield prediction sets composed of multiple disconnected subintervals, which are difficult to interpret. In this paper, we propose SCD-split, which incorporates smoothing operations into the CP framework. Such smoothing operations potentially help merge the subintervals, thus leading to interpretable prediction sets. Experimental results on both synthetic and real-world datasets demonstrate that SCD-split balances the interval length and the number of disconnected subintervals. Theoretically, under specific conditions, SCD-split provably reduces the number of disconnected subintervals while maintaining comparable coverage guarantees and interval length compared with CD-split.

## 1 Introduction

Machine learning models have achieved remarkable success across numerous applications, including large language models (Chang et al., 2024), medical diagnosis (Marcinkevičs et al., 2022), and investment (Papasotiriou et al., 2024). Despite the impressive performance and widespread adoption, they are often sensitive to noise, model misspecification, and inference errors (Abdar et al., 2021), which undermine the prediction reliability and thus limit their practical applicability in high-stakes scenarios (Nguyen et al., 2015; Hein et al., 2019; Martino et al., 2023; Huang et al., 2025a). Consequently, this has raised interest in developing rigorous methods for *uncertainty quantification* to enhance the trustworthiness of machine learning outputs (Guo et al., 2017; Kristiadi et al., 2020).

Among various uncertainty quantification approaches, conformal prediction (CP) has emerged as a powerful and versatile framework that provides statistically rigorous uncertainty guarantees under mild assumptions (Vovk et al., 2005; Romano et al., 2019a; Angelopoulos & Bates, 2021). CP wraps around black-box predictive models and outputs prediction sets whose validity is ensured by data exchangeability, without requiring knowledge of the underlying data distribution. Owing to its strong theoretical guarantees and model-agnostic flexibility, conformal prediction has demonstrated promising and growing applicability across a variety of fields, including drug discovery (Laghuvarapu et al., 2023), large language model (Gui et al., 2024), and health care (Eghbali et al., 2024).

Beyond guaranteeing the coverage, conformal prediction is expected to produce prediction sets with smaller lengths to enable more informative uncertainty quantification in practice. Consequently, many approaches have been proposed to improve length efficiency under the conformal prediction framework (Romano et al., 2019c; Teng et al., 2023; Izbicki et al., 2019). Among them, CD-split (Izbicki et al., 2021) stands out due to its strong performance in improving length efficiency, which uses the conditional density estimation as the conformity score. It approximately achieves prediction sets with minimal Lebesgue measure while maintaining valid coverage when the conditional density estimation is accurate (Izbicki et al., 2019).

Despite the strong theoretical properties of CD-split, several practical challenges arise when it is applied to real-world scenarios. When the conditional distribution is complex or highly multimodal, the prediction sets generated by CD-split often consist of many small disconnected intervals[1] (see Figure 1). The lack of connectivity makes the prediction sets difficult to interpret and thus limits their usefulness in practical tasks where clear and concise predictions are preferred.

---

[1]We here restrict our discussions to the regression tasks (See Appendix B for more discussions).

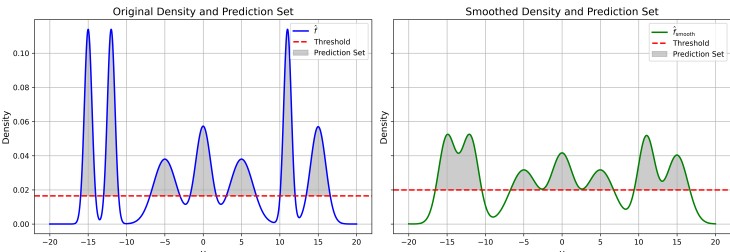

Figure 1: Illustration of Fourier smoothing on a synthetic multimodal distribution. The original density (left) contains seven sharp peaks, resulting in prediction sets composed of multiple disconnected intervals. After applying a smoothing technique (right), the number of intervals is reduced to three with a mild increase in total length, improving the interpretability of the prediction sets.

In this paper, we propose *SCD-split* to address the above challenges. SCD-split explicitly focuses on the interpretability of prediction sets. Specifically, interpretability measures how clearly and intuitively the prediction sets convey information to users, which is quantified using both interval length and the number of disjoint intervals (connectivity). Within SCD-split, users first specify a desired number of disjoint intervals that they regard as appropriate for interpretation. To meet this requirement, SCD-split applies a smoothing technique to the fitted conditional density function before constructing the prediction sets. This smoothing step reduces unnecessary peaks in the estimated density function and corrects distortions caused by complex noise or overfitting, making the density estimation more stable and meaningful. We further use the validation process to tune the smoothing parameter so that the final prediction set conforms to the specified number of intervals as closely as possible. As a result, the final prediction sets contain disconnected intervals matching users' desired number, making them easier to interpret while maintaining the coverage guarantee. While smoothing has long been a standard tool, this is the first work to introduce it into conformal prediction frameworks to directly regulate the connectivity of prediction sets, thereby providing a novel and principled approach to shape their structure. We refer to Figure 1 for a visual illustration.

**Theoretically**, we first prove in Theorem 4.1 that the proposed smoothing procedure preserves the marginal coverage guarantee of conformal prediction. Second, under general conditions, we establish that smoothing techniques lead to controlled behaviors: the length of the prediction sets admits a provable upper bound (Theorem 4.4), and the number of disconnected intervals does not increase after smoothing (Theorem 4.2). Third, we prove that smoothing strictly reduces the number of intervals under specific structural assumptions—such as narrow-valley double peaks (Theorem 4.3), thereby improving the interpretability of the prediction sets without sacrificing coverage. **Empirically**, we evaluate our method on both synthetic and real-world datasets in Section 5. The results show that our method achieves a favorable trade-off between interval length and the number of intervals while maintaining validity, particularly under complex and multimodal distributions. Such balance leads to a notable improvement in the interpretability of the prediction sets.

In many practical problems, controlling the number of disjoint intervals in prediction sets is essential for making the results interpretable and actionable. We briefly present two motivating examples:

**Health Care.** In medical prognosis, when a disease's course is highly uncertain, doctors often face diseases whose future course may branch into a few qualitatively different trajectories—for example, a fast-progressing fatal path and a long-term recovery path. For treatment planning and patient counseling, it is important to express these distinct possibilities through a manageable number of separate time ranges, rather than a single broad interval that mixes them together or an excessive number of small intervals that are difficult to interpret. Our method allows physicians to directly specify the desired number of disjoint intervals so that the resulting conformal prediction sets provide interpretable and clinically actionable uncertainty quantification.

**Finance.** In stock price forecasting, market conditions may be highly uncertain, and investors may face two qualitatively different outcomes: a strong upward movement or a significant decline. In such cases, they often wish to know two separate ranges with higher accuracy—one indicating how high the price may rise if the market strengthens, and another showing how low it may fall if conditions worsen. Our method allows investors to pre-specify the desired number of disjoint intervals, so that the resulting conformal prediction set clearly distinguishes these up-side and down-side scenarios and provides more concrete guidance for trading and risk management.

## 2 RELATED WORK

**Conformal Prediction.** Conformal prediction (Vovk et al., 2005; Shafer & Vovk, 2008; Barber et al., 2020) is a statistical framework that turns black-box model outputs into predictive intervals. It offers several desirable properties, including distribution-free, non-asymptotic guarantees and a user-friendly implementation (Angelopoulos & Bates, 2021). Existing research on conformal prediction mainly focuses on two aspects: interval length and coverage guarantee. Interval length is an important metric measuring the performance of conformal prediction methods (Teng et al., 2023; 2021; Angelopoulos et al., 2020; Zhou & Sesia, 2024). To minimize the predicted interval length, researchers try to build adaptive prediction intervals (Romano et al., 2019b; Lu, 2024), modify non-conformity scores (Izbicki et al., 2019; Wang & Qiao, 2025) or regard interval length as the optimization objective (Stutz et al., 2022; Kiyani et al., 2024; Bars & Humbert, 2025). For coverage guarantee, numerous works focus on improving the conditional coverage (Romano et al., 2019a; Gibbs et al., 2024; Plassier et al., 2025). Unfortunately, conditional coverage holds only on some special distributions (Vovk, 2012; Lei & Wasserman, 2014; Barber et al., 2020). Therefore, work on conditional coverage can be roughly split into two branches: (a) *local coverage* (Lei & Wasserman, 2014; Barber et al., 2020; Guan, 2023) controls the conditional coverage in a pre-selected space; (b) *asymptotic coverage* (Lei et al., 2018; Izbicki et al., 2019; Sesia & Romano, 2021) establishes conditional coverage guarantees that hold asymptotically as the sample size tends to infinity.

**Conformal Prediction and Interpretability.** Conformal prediction has been used to enhance the interpretability of the model (Johansson et al., 2018; Sanchez-Martin et al., 2024; Qian et al., 2024) in various fields that need interpretability and reliability, *e.g.*, medicine (Lu et al., 2022; Hirsch & Goldberger, 2024; Huang et al., 2025b) and finance (Zaffran et al., 2022). However, the interpretability of conformal prediction techniques is still underexplored, *e.g.*, confidence intervals with multiple disconnected intervals may potentially influence the interpretability of conformal prediction.

**Smoothing.** Smoothing methods have been used in various fields, *e.g.*, computer vision (Wang et al., 2022), statistics (Chacón et al., 2013; Ho & Walker, 2020) and numerical analysis (Pandey & Anand, 2020). In this paper, we mainly focus on Fourier smoothing and randomized smoothing. Fourier smoothing uses different frequency-domain filters, *e.g.*, ideal low-pass filter (ILPF) (Jeon et al., 2024), Gaussian low-pass filter (GLPF) (Mehrabkhani, 2019; 2022), Butterworth low-pass filter (BLPF) (Xiao & Bo, 2025) and window functions (Ohamouddou et al., 2025). Randomized smoothing has been used for constructing adversarial robustness classifiers (Cohen et al., 2019; Teng et al., 2020). Gendler et al. (2019); Yan et al. (2024) introduce randomized smoothing into conformal prediction and propose randomized smoothing conformal prediction (RSCP), which is a robust conformal prediction framework under adversaries. Unlike RSCP which applies randomized smoothing at the input level to improve robustness against adversarial perturbations, our method smooths the estimated conditional density function to improve interpretability.

## 3 METHODOLOGY

In this section, we propose our SCD-split framework to improve the interpretability of prediction sets constructed from conditional density estimators. We first review the classical split conformal prediction and its extension to density-based methods in Section 3.1. Then, we introduce a Fourier-based smoothing technique to regularize the estimated conditional densities in Section 3.2. Finally, we summarize the complete SCD-split procedure in Section 3.3.

### 3.1 CONFORMAL PREDICTION BASED ON CONDITIONAL DENSITY ESTIMATOR

**Split conformal prediction** is a commonly adopted method in conformal prediction, which constructs valid prediction sets through a data-splitting procedure. Specifically, the dataset is divided into two disjoint subsets: training set $\mathcal{D}_{tr}$ and calibration set $\mathcal{D}_{ca}$. A predictor $\hat{f}$ is trained on $\mathcal{D}_{tr}$, and non-conformity scores $V(X_i, Y_i)$ are computed on $\mathcal{D}_{ca}$ symmetrically. This validity property relies on a mild assumption about the data: the exchangeability of the data pairs in Assumption 3.1.

**Assumption 3.1** (Exchangeability). *Define $\{Z_i\}_{i=1}^n$, as the data pairs, then $Z_i$ are exchangeable if arbitrary permutation follows the same distribution, i.e.,*

$$(Z_1, \ldots, Z_n) \stackrel{d}{=} (Z_{\pi(1)}, \ldots, Z_{\pi(n)}), \tag{1}$$

*with arbitrary permutation $\pi$ over $\{1, \ldots, n\}$.*

This setup ensures that the non-conformity score $V_{n+1}$ for a new test point is exchangeable with the scores in $\mathcal{D}_{\mathrm{ca}}$, which in turn implies that the rank of $V_{n+1}$ among $V_1, V_2, \ldots, V_{n+1}$ is uniformly distributed. Consequently, a valid prediction set can be formed using a quantile-based threshold:

$$\mathcal{C}_{1-\alpha}(X_{n+1}) = \{y : V(X_{n+1}, y) \leq \mathrm{Quantile}(1 - \alpha; \{V_i\}_{i \in \mathcal{I}_{\mathrm{ca}}} \cup \{+\infty\})\}, \qquad (2)$$

where $\mathcal{I}_{\mathrm{ca}}$ denotes the index set corresponding to the calibration set $\mathcal{D}_{\mathrm{ca}}$. This approach guarantees marginal coverage for prediction sets:

$$\mathbb{P}(Y_{n+1} \in \mathcal{C}_{1-\alpha}(X_{n+1})) \geq 1 - \alpha. \qquad (3)$$

**CD-split.** Within the framework of split conformal prediction, a notable class of methods constructs prediction sets via conditional density estimation (Izbicki et al., 2021; 2019). These methods train a conditional density estimator $\hat{f}(y \mid x)$ on the training set $\mathcal{D}_{\mathrm{tr}}$ and compute conformity scores directly as $\{\hat{f}(y_i \mid x_i), i \in \mathcal{I}_{\mathrm{ca}} \cup \{n + 1\}\}$ using the calibration set $\mathcal{D}_{\mathrm{ca}}$ and the test point. Given the exchangeability of the data points $\{Z_i\}_{i=1}^{n+1}$, these conformity scores are also exchangeable, enabling the construction of valid prediction sets under the split conformal prediction framework.

Among them, CD-split is a representative approach that exemplifies this strategy (Izbicki et al., 2021). Building on this framework, CD-split further clusters the input space and, when constructing prediction sets, uses only the calibration data that are similar to each test point. Through these mechanisms, CD-split constructs prediction sets that asymptotically converge to the oracle highest predictive density set (Proposition C.1 in Izbicki et al. (2021)). This property enables CD-split to produce smaller prediction sets compared to interval-based methods when the conditional density estimation performs well, thereby yielding improvements in efficiency. However, such mechanism may produce disconnected prediction sets under multimodal distributions, which hinders interpretability and poses challenges in practical applications. We refer to Section 5 for more details.

### 3.2 SMOOTHING TECHNIQUE

This section introduces the basics of smoothing techniques. Despite the efficiency advantages of CD-split, it potentially yields prediction sets with multiple disconnected intervals when the estimated conditional density is complex or highly multimodal. To address this issue, we introduce a Fourier-based smoothing technique which regularizes the estimated density function. Specifically, the Fourier transform in Definition 3.1 utilizes the powerful frequency-domain representation of functions to reduce noise and high-frequency oscillations in the estimated conditional density.

**Definition 3.1** (Fourier Smoothing for Conditional Density Estimation)**.** *Let $\hat{f}(y \mid x)$ be an estimated conditional density. We apply Fourier smoothing in the response variable $y$ as follows: First compute the Fourier transform of $\hat{f}(y \mid x)$ with respect to $y$: $\mathcal{F}_y[\hat{f}](w \mid x) = \int_{-\infty}^{\infty} \hat{f}(y \mid x) e^{-2\pi i y w} dy$. Then multiply this transform by a Gaussian low-pass filter $H_\sigma(w) = e^{-2\pi^2 \sigma^2 w^2}$, where the smoothing parameter $\sigma > 0$ controls the strength of smoothing. The smoothed spectrum is $\mathcal{F}_y[\hat{f}]^{\mathrm{FS}}(w \mid x) = \mathcal{F}_y[\hat{f}](w \mid x) H_\sigma(w)$, and the smoothed conditional density is obtained by the inverse transform*

$$\tilde{f}^{\mathrm{FS}}(y \mid x) = \int_{-\infty}^{\infty} \mathcal{F}_y[\hat{f}]^{\mathrm{FS}}(w \mid x) e^{2\pi i y w} dw. \qquad (4)$$

The key insight is that sharp variations or spurious peaks in the density function correspond to high-frequency components in its spectral representation. By applying Fourier smoothing, we reduce the number of local modes in $\hat{f}(y \mid x)$, especially those arising from estimation noise. This has a direct impact on the structure of the prediction sets generated by CD-split: the number of disjoint intervals is reduced, and the resulting sets are more concise and easier to present and interpret, while still preserving valid coverage guarantees.

### 3.3 SCD-SPLIT ALGORITHM

The proposed algorithm integrates a smoothing technique into the CD-split framework, aiming to enhance the interpretability of prediction sets while preserving CD-split's desirable theoretical properties, such as local conditional coverage. Each step of the pipeline is described below in detail:

---

**Algorithm 1** SCD-split

---

**Input:** Dataset $\mathcal{D} = \{(\mathbf{X}_i, Y_i)\}_{i=1}^n$, confidence level $1 - \alpha \in (0,1)$, training algorithm $\mathcal{T}$ for conditional density, smoothing operator $s_\sigma(\cdot)$, candidate grid $\Sigma$ for $\sigma$, prespecified target number of intervals $K_{\text{target}}$.

1: Randomly split $\mathcal{D}$ into training $\mathcal{D}_{\text{tr}}$, validation $\mathcal{D}_{\text{val}}$, and calibration $\mathcal{D}_{\text{ca}}$.
2: Fit $\hat{f} = \mathcal{T}(\mathcal{D}_{\text{tr}})$ where $\hat{f}(Y_i \mid \mathbf{X}_i)$ is the estimated conditional density;
3: **for** each $\sigma \in \Sigma$ **do**
4: $\quad$ Smooth the estimator: $\tilde{f}_\sigma^{\text{FS}}(y \mid \mathbf{x}) \leftarrow s_\sigma(\hat{f}(y \mid \mathbf{x}))$.
5: $\quad$ Compute a partition $\mathcal{A}_\sigma$ of $\mathcal{X}$ by clustering training samples via profile distance (Def. C.3).
6: $\quad$ For each cell $a \in \mathcal{A}_\sigma$, form $\mathrm{U}_\sigma(a) = \{\tilde{f}_\sigma^{\text{FS}}(Y_i \mid \mathbf{X}_i) : (\mathbf{X}_i, Y_i) \in \mathcal{D}_{\text{ca}}, \ \mathbf{X}_i \in a\}$ and compute threshold $t_\sigma^S(a) = \text{Quantile}(\alpha; \mathrm{U}_\sigma(a))$.
7: $\quad$ For each $(\mathbf{X}_j, Y_j) \in \mathcal{D}_{\text{val}}$, find its cell $a_j \in \mathcal{A}_\sigma$, construct $\mathcal{C}_\sigma^S(\mathbf{X}_j) = \{y : \tilde{f}_\sigma^{\text{FS}}(y \mid \mathbf{X}_j) \geq t_\sigma^S(a_j)\}$, and record the number of disjoint intervals $N_\sigma(\mathbf{X}_j)$.
8: $\quad$ Compute $R(\sigma) = \left| \frac{1}{|\mathcal{D}_{\text{val}}|} \sum_{(\mathbf{X}_j, Y_j) \in \mathcal{D}_{\text{val}}} N_\sigma(\mathbf{X}_j) - K_{\text{target}} \right|$.
9: **end for**
10: Select $\hat{\sigma} \in \arg\min_{\sigma \in \Sigma} R(\sigma)$; set $\tilde{f}^{\text{FS}} \leftarrow \tilde{f}_{\hat{\sigma}}^{\text{FS}}$ and $\mathcal{A} \leftarrow \mathcal{A}_{\hat{\sigma}}$.
11: Find the partition $a(\mathbf{X}_{n+1}) \in \mathcal{A}$ with $\mathbf{X}_{n+1} \in a(\mathbf{X}_{n+1})$;
12: Form the set $\mathrm{U}(\mathbf{X}_{n+1}, \mathcal{D}_{\text{ca}}) = \{\tilde{f}^{\text{FS}}(Y_i \mid \mathbf{X}_i) : (\mathbf{X}_i, Y_i) \in \mathcal{D}_{\text{ca}}, \ \mathbf{X}_i \in a(\mathbf{X}_{n+1})\}$;
13: Compute $t^S = \text{Quantile}(\alpha; \mathrm{U}(\mathbf{X}_{n+1}, \mathcal{D}_{\text{ca}}))$;
**Output:** Prediction set $\mathcal{C}_{1-\alpha}^S(\mathbf{X}_{n+1}) = \{y : \tilde{f}^{\text{FS}}(y \mid \mathbf{X}_{n+1}) \geq t^S\}$.

---

**Dataset.** Consider an exchangeable dataset $\mathcal{D} = \{(\mathbf{X}_i, Y_i)\}_{i=1}^n$. We randomly split $\mathcal{D}$ into three parts: a training set $\mathcal{D}_{\text{tr}}$ for fitting the conditional density, a validation set $\mathcal{D}_{\text{val}}$ for tuning the smoothing parameter, and a calibration set $\mathcal{D}_{\text{ca}}$ for constructing the final prediction sets. We further assume that the test pair $(\mathbf{X}_{n+1}, Y_{n+1})$ is exchangeable with $\mathcal{D}$.

**Choice of target interval number.** The target interval number $K_{\text{target}}$ is predetermined by the user based on domain knowledge and the specific requirements of the application. This user-specified quantity serves as a way to incorporate prior understanding of the problem's structural characteristics into the modeling process, ensuring that the resulting prediction sets are interpretable.

**Training process.** We utilize the machine learning model, such as random forest or neural networks, to train a model via the training set $\mathcal{D}_{\text{tr}}$.

**Choosing the smoothing parameter $\sigma$.** We select the smoothing parameter $\sigma$ by evaluating all candidate values on the validation set and choosing the one whose prediction sets yield an average number of disjoint intervals closest to the user-specified target $K_{\text{target}}$.

*First*, for each candidate value $\sigma$, we smooth the estimated density $\hat{f}$ by applying the Fourier smoothing operator $s_\sigma$, obtaining $\tilde{f}_\sigma^{\text{FS}} = s_\sigma(\hat{f})$. Based on $\tilde{f}_\sigma^{\text{FS}}$, we compute conditional CDF profiles and perform $k$-means++ clustering on the training covariates based on the profile distance (Definition C.3 in (Izbicki et al., 2021)) to form a partition $\mathcal{A}_\sigma$ of the input space $\mathcal{X}$.

*Second*, we use the calibration set $\mathcal{D}_{\text{ca}}$ to construct provisional conformal thresholds for each cell of $\mathcal{A}_\sigma$. These thresholds allow us to form prediction sets on the validation set. Then, we record the number of disjoint intervals for every validation point. Finally, we compute the difference between the average number of intervals and the user-specified target $K_{\text{target}}$, and select the $\sigma$ whose prediction sets best match this target. The chosen $\sigma$ determines both the final smoothed estimator $\tilde{f}^{\text{FS}}$ and the final partition $\mathcal{A}$. We further discuss the choice of loss function in Appendix D.2.

**Calibration and testing process.** For a new test point $\mathbf{X}_{n+1}$, we locate the cell $a(\mathbf{X}_{n+1}) \in \mathcal{A}$ containing it and form the final conformal prediction set $\mathcal{C}_{1-\alpha}^S(\mathbf{X}_{n+1})$ using the smoothed density $\tilde{f}^{\text{FS}}$ and the corresponding calibrated threshold.

## 4 THEORETICAL GUARANTEE

This section presents theoretical guarantees for SCD-split. Specifically, Theorem 4.1 establishes the finite-sample coverage guarantee of prediction sets. Theorem 4.2 demonstrates that the smoothing

operation does not increase the number of disconnected intervals, thereby improving connectivity. Theorem 4.3 provides a special structural case that the number of disconnected intervals decreases after smoothing. Lastly, Theorem 4.4 provides an upper bound on the length increase of the predicted intervals produced by the SCD-split algorithm compared with CD-split.

**Theorem 4.1** (Coverage Preservation). *Let $\alpha \in (0, 1)$ and assume the data pairs $\{(\mathbf{X}_i, Y_i)\}_{i=1}^{n+1}$ are exchangeable. For a new data point $\mathbf{X}_{n+1}$, let $\mathcal{C}_{1-\alpha}^S(\mathbf{X}_{n+1})$ denote the prediction sets of $\mathbf{X}_{n+1}$ predicted by the SCD-split algorithm. Then*

$$\mathbb{P}\big(Y_{n+1} \in \mathcal{C}_{1-\alpha}^S(\mathbf{X}_{n+1})\big) \geq 1 - \alpha. \tag{5}$$

The intuition for theorem 4.1 is that the smoothing operation preserves the symmetry property of the score function with respect to the calibration and test data. Consequently, the exchangeability of the data pairs $\{(\mathbf{X}_i, Y_i)\}_{i=1}^{n+1}$ naturally induces the exchangeability of the conformity scores. This key property ensures that SCD-split retains the finite-sample coverage guarantee. The detailed proof is provided in Appendix C.2.

**Theorem 4.2** (Non-increasing interval count). *Suppose $f : \mathbb{R} \to \mathbb{R}$ is bounded, measurable, and differentiable, and denote $\mathcal{B} = \{t : f(x) = t, f'(x) \neq 0\}$. Let $\tilde{f}^{FS}$ be the function after applying Fourier smoothing to function $f$. For all $t \in \mathcal{B}$ such that $A_t := \{x : f(x) = t, f'(x) \neq 0\}$ is finite, i.e., $\#A_t < \infty$, it holds that*

$$\#B_t := \#\{x : \tilde{f}^{FS}(x) = t\} \leq \#A_t, \tag{6}$$

*where $\#$ denotes the cardinality of the set. Namely, the number of disconnected intervals of $B_t$ is less than or equal to that of $A_t$.*

Intuitively, the number of disconnected intervals would not increase because Fourier smoothing merges small oscillations. Theorem 4.2 indicates that the number of intervals predicted by SCD-split does not exceed those before smoothing, thereby enhancing interpretability. We assume here that the threshold $t$ remains unchanged for simplicity, as the smoothing operation does not significantly change its value. The detailed proof is provided in Appendix C.3.

**Theorem 4.3** (Strict merging under narrow-valley structure). *Assume $f : \mathbb{R} \to \mathbb{R}$ to be bounded, measurable, and differentiable, and let $\tilde{f}^{FS}$ denote its Fourier smoothed version. For a fixed threshold $t \in \mathcal{B}$, suppose there exist two adjacent intervals $(a_1, b_1)$ and $(a_2, b_2)$, both subsets of $\{x : f(x) \geq t\}$, such that the valley region $(b_1, a_2)$ between them satisfies:*

- *$f(x) \leq t - \varepsilon$ for all $x \in (b_1, a_2)$, for some $\varepsilon > 0$;*

- *The gap width $\delta := a_2 - b_1$ satisfies $\int_{|u| \geq \delta/2} \phi_\sigma(u)\, du \geq \frac{\varepsilon}{\|f\|_\infty}$, where $\phi_\sigma$ is the Gaussian kernel used in the convolution.*

*Then after smoothing, the number of intervals strictly decreases:*

$$\#\{x : \tilde{f}^{FS}(x) \geq t\} < \#\{x : f(x) \geq t\}. \tag{7}$$

The intuition for theorem 4.3 is that when two high regions of the function are separated by a narrow and shallow valley, the Gaussian kernel convolution has sufficient smoothing power to "fill in" the valley, effectively merging previously disconnected regions into a single connected interval. The detailed proofs are provided in Appendix C.4. We also provide another special case in Appendix C.6.

**Theorem 4.4** (Interval length bound). *Define $\sigma$ as the smoothing factor of Fourier smoothing, $N$ as the number of disconnected intervals predicted by CD-split, the original estimated conditional density function as $\hat{f}(y \mid \mathbf{X})$ and the smoothed conditional density function as $\tilde{f}^{FS}(y \mid \mathbf{X})$. Assume $\hat{f}$ and $\tilde{f}^{FS}$ are L-Lipschitz and satisfy $|f(y_1 \mid \mathbf{X}) - f(y_2 \mid \mathbf{X})| \geq M|y_1 - y_2|, \quad f \in \{\hat{f}, \tilde{f}^{FS}\}$. Then the difference between the original predicted interval length $l$ and the smoothed predicted interval length $\tilde{l}$ satisfies*

$$|\tilde{l} - l| \leq \frac{4NL\sigma}{M} \sqrt{\frac{2}{\pi}}. \tag{8}$$

The intuition for Theorem 4.4 is that the uniform bound on the pointwise deviation between the original and smoothed conditional densities leads to a bounded shift in the empirical quantile thresholds which determine the endpoints of the intervals. Consequently, the reduction in the number of intervals predicted by SCD-split given by theorem 4.2 does not result in excessively long intervals, thus preserving interpretability. We provide a detailed statement and proof in Appendix C.5.

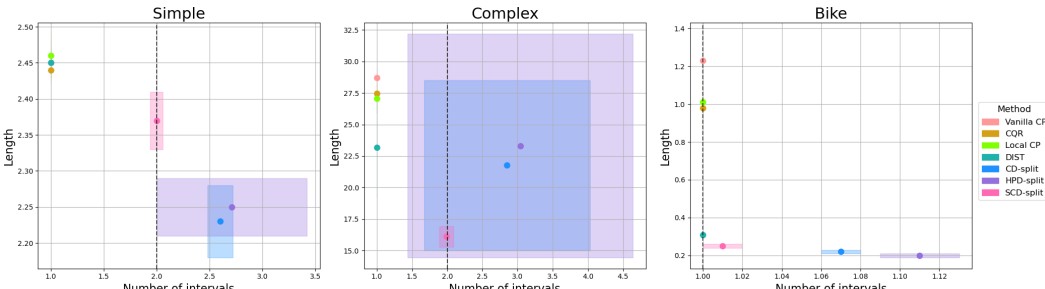

Figure 2: Length vs Number of intervals under complex synthetic and real-world data settings. Each rectangle size shows the standard deviation around the mean. Points closer to the black dashed vertical line on the $x$-axis (the target number of intervals) and lower on the $y$-axis (shorter length) indicate better performance. Our proposed SCD-split method consistently reaches the target number of intervals while maintaining shorter lengths across all tasks, demonstrating strong overall performance. We defer more related details to Appendix D.

## 5 EXPERIMENTS

We conduct experiments on synthetic and real-world datasets, mainly to show that SCD-split is (a) effective, *i.e.*, it constructs valid prediction sets with empirical coverage larger than $1 - \alpha$, (b) efficient, *i.e.*, it constructs prediction sets with relatively short interval length, and (c) interpretable, *i.e.*, it constructs prediction sets with interval number close to target number.

### 5.1 SETUP

**Datasets:** We evaluate our method on both *synthetic* and *real-world* datasets. The synthetic datasets include two types: a simple multimodal distribution generated by mixing three Gaussian components with identical variances, and a more complex multimodal distribution formed by mixing multiple Gaussians with varying means and variances. For real-world evaluation, we use several standard datasets commonly adopted in conformal prediction studies, such as `bio` and `bike`, covering diverse application domains and distributional characteristics.

**Baselines**: We compare our method against two categories of baselines. The first category consists of standard split conformal prediction methods, including vanilla conformal prediction (CP) (Vovk et al., 2005), conformalized quantile regression (CQR) (Romano et al., 2019c), and local conformal prediction (LCP) (Guan, 2023). The second category includes methods based on conditional density estimation, specifically dist, CD-split, and HPD-split (Izbicki et al., 2019; 2021). To ensure a fair comparison across all methods, we uniformly use random forests as the underlying predictive model.

**Evaluation metrics**: We evaluate all methods using three metrics. The first metric is empirical coverage, which measures the proportion of true responses captured by the prediction sets; a valid method should achieve coverage greater than or equal to $1 - \alpha$. The second metric is the interval length, where a shorter length indicates more precise predictions. The third metric is the number of disjoint subintervals, with values closer to the target number indicating better interpretability.

**Synthetic data.** We generate synthetic data to evaluate performance across varying levels of complexity. The covariate vector $X = (X_1, \ldots, X_d)$ is sampled *i.i.d.* from Unif$(-5, 5)$ and standardized. The response variable $Y \mid X$ follows a flexible multi-modal mixture model:

$$Y \mid X \sim \sum_{k=1}^{K} \frac{\exp(X^\top \beta_k)}{\sum_{j=1}^{K} \exp(X^\top \beta_j)} \mathcal{N}(\mu_{\text{base},k} + X^\top \gamma_k, \sigma_k^2), \tag{9}$$

where the parameters are constructed to control the number, location, and shape of the modes. This setup allows us to evaluate both simple and complex structures under a unified framework.

**Real-world data.** We conduct experiments on several real-world datasets commonly used in the conformal prediction literature (Romano et al., 2019a; Teng et al., 2023), including the bike sharing dataset (`bike`) (Fanaee-T, 2013) and physicochemical properties of protein tertiary structure dataset (`bio`) (Rana, 2013). On these datasets, the fitted conditional density estimates are both multi-modal. We defer more related details to Appendix D.

Table 1: Different smoothing parameters on synthetic complex dataset

| Method / $\sigma$ | Coverage (%) | Length | Number of Intervals |
|---|---|---|---|
| CD-split ($\sigma = 0$) | $91.06 \pm 3.55$ | $21.76 \pm 6.74$ | $2.85 \pm 1.18$ |
| SCD-split ($\sigma = 1$) | $89.38 \pm 0.92$ | $16.20 \pm 0.80$ | $2.51 \pm 0.21$ |
| SCD-split ($\sigma = 1.5$) | $89.23 \pm 0.77$ | $16.11 \pm 0.68$ | $1.99 \pm 0.01$ |
| SCD-split ($\sigma = 2$) | $89.39 \pm 0.85$ | $16.53 \pm 0.36$ | $1.74 \pm 0.06$ |
| SCD-split ($\sigma = 5$) | $89.42 \pm 0.86$ | $19.78 \pm 1.05$ | $1.18 \pm 0.02$ |
| SCD-split ($\sigma = 10$) | $89.47 \pm 1.00$ | $22.53 \pm 1.10$ | $1.00 \pm 0.00$ |

## 5.2 RESULT AND DISCUSSION

**Validity.** We summarize the empirical coverage in Table 2 and Table 3. The results show that the empirical coverage achieved by all methods matches the theoretical target $1 - \alpha$, demonstrating the effectiveness of the proposed procedures.

**Efficiency.** We summarize the interval lengths in Table 2, Table 3 and Figure 2. We evaluate the efficiency of each method by measuring the average length of the prediction sets. On both synthetic and real-world datasets, we find that methods based on conditional density estimation generally produce much shorter prediction intervals than standard conformal prediction methods. This is because density-based methods allow disconnected prediction sets, which makes it possible to include only the regions with high estimated probability and avoid unnecessary coverage in low-density areas. After applying the smoothing technique, we observe two different behaviors depending on the complexity of the data. When the conditional distribution is not very complex, smoothing slightly increases the interval length due to the regularization effect, but this increase is small and acceptable. However, when the distribution is highly multimodal or the data is noisy, smoothing helps remove spurious modes and reduces the influence of noise in the estimated densities. As a result, the prediction sets become shorter especially in complex settings or real-world data with large noise, which improves efficiency while maintaining valid coverage.

**Connectivity and Interpretability.** We evaluate connectivity by reporting the number of intervals in Table 2, Table 3, Figure 2. Different from classical conformal prediction methods which usually produce a single connected interval, density-based approaches such as CD-split and HPD-split often generate prediction sets with many disconnected components. Our results imply that applying smoothing operations allows SCD-split to accurately approach the user-specified target number of intervals, which leads to prediction sets that are both faithful to the desired structure and easier to interpret. *Furthermore*, we assess interpretability by jointly considering how close the number of intervals is to the target and how small the total interval length remains. SCD-split consistently achieves a favorable trade-off between these two metrics: compared with existing methods, it brings the number of intervals closer to the target while keeping lengths competitive.

**Ablation on the smoothing parameter $\sigma$.** We investigate the impact of the smoothing parameter $\sigma$ on the performance of the proposed method in Table 1. The table reports test results obtained by directly fixing $\sigma$ in advance and skipping the validation process, in order to directly demonstrate how different $\sigma$ values affect the prediction sets on the test data. When $\sigma$ is close to zero, the smoothing effect is negligible, and the results are nearly identical to those of the original CD-split method. As $\sigma$ increases, the smoothing effect gradually strengthens, effectively removing spurious modes in the estimated density. When $\sigma$ becomes very large (*e.g.*, $\sigma = 10$), our method degenerates to producing a single, broad prediction interval. The number of disjoint intervals decreases smoothly as $\sigma$ grows. While in practice increasing the smoothing parameter does not always guarantee such a strictly monotone decrease, we find empirically that this pattern holds in most cases. Therefore, by appropriately setting the range of candidate $\sigma$ values and applying our validation process, we make the resulting prediction sets match the pre-specified target number of intervals, achieving a desirable balance between efficiency and interpretability.

**Ablation on smoothing techniques.** Table 4 presents the experimental results obtained with different smoothing techniques. The experimental results imply that several smoothing techniques perform well under our framework and achieve prediction sets whose number of intervals is close to the target number. This demonstrates that our framework is general, allowing users to choose a smoothing technique suited to the characteristics of their specific application to obtain more interpretable

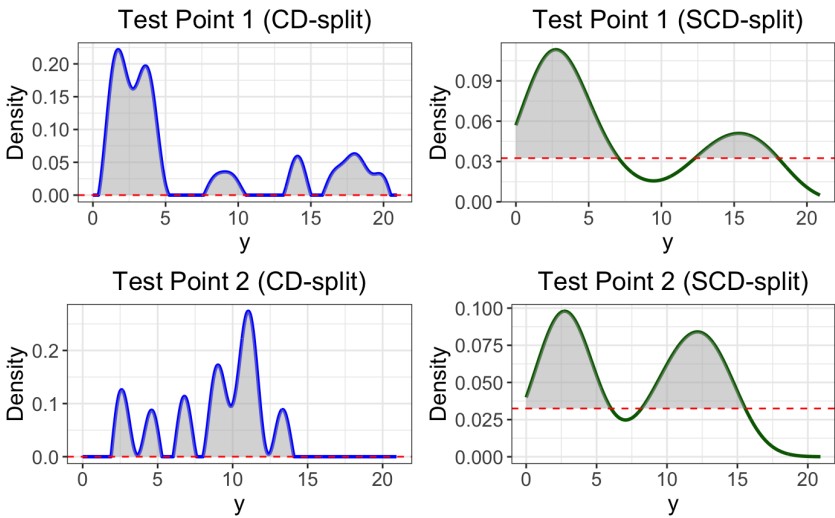

Figure 3: Illustration of prediction set construction on the first two test points from the `Bio` dataset. In each panel, the solid curve represents the estimated conditional density ($\hat{f}(y \mid x)$), the dashed red line indicates the threshold, and the shaded region marks the resulting prediction sets. As shown in the two left panels, $\hat{f}(y \mid x)$ equals zero over large regions, driving the threshold to zero and causing CD-split to return the entire output domain of $y$ as the prediction set. By contrast, the right two panels demonstrate how SCD-split produces a smoother density function, which breaks the ties and restores coverage at the desired level $1 - \alpha$, thereby yielding more interpretable prediction sets.

prediction sets. Moreover, we observe empirically that Fourier smoothing yields prediction sets with smaller average lengths and thus better interpretability compared with other smoothing techniques when the smoothing parameter is chosen properly. Therefore, we choose Fourier smoothing as the primary technique in our framework.

**Extreme case.** We observe that on real-world datasets, conditional density estimation may perform poorly, and both CD-split and HPD-split are highly sensitive to such estimation errors. For instance, in `Bio` dataset, the experimental result in Table 3 implies that these errors can lead to overly large prediction sets and poor interpretability. The reason is that when the estimated density is poor, a large portion of calibration responses $y_i$ may fall in regions where $\hat{f}(y_i \mid x_i) = 0$, resulting in conformity scores collapsing to zero. If the proportion of such ties exceeds $\alpha$, the resulting prediction sets may degenerate into the entire output domain, yielding empirical coverage close to 100%. A common solution is to assign a small random value to break these ties and restore the coverage of $1 - \alpha$. However, this typically leads to poorer interpretability. To address this issue, our methods apply smoothing operation to $\hat{f}$, which effectively merges spurious zero-density regions into surrounding modes. This process breaks excessive ties and restores a more meaningful distribution of conformity scores, allowing the coverage to return to the target level of $1 - \alpha$ while keeping the prediction sets informative. In this case, smoothing may increase the interval number, which appears at first sight to conflict with our theoretical guarantee. However, this apparent discrepancy is explained by the violation of an assumption in Theorem 4.2, namely that the set $A_t := \{x : f(x) = t, f'(x) \neq 0\}$ is finite for all $t \in \mathcal{B}$. We refer to Figure 3 for illustration.

## 6 CONCLUSION

In this work, we propose a smoothing-based framework to enhance the interpretability of prediction sets, particularly for methods based on conditional density estimation. Theoretically, we show that smoothing preserves desirable properties of the prediction sets. Empirically, our method achieves a favorable trade-off between interval length and number across both synthetic and real-world datasets, thereby substantially improving the interpretability of the prediction sets. Consequently, this paper suggests that smoothing serves as a practical enhancement for conformal prediction methods based on conditional density estimation. It might be interesting to explore task-specific smoothing techniques that adaptively balance interval length and connectivity for interpretability in future work.

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

# Appendix

In Section A, we provide some supplementary statements in this paper. In Section B, we provide several additional discussions to further contextualize and clarify the practical scope and our contributions. In Section C, we provide some detailed proofs omitted in the main context. Specifically, in Section C.1, we provide definitions of essential concepts that are repeatedly invoked in the proofs that follow. In Section C.2, we rigorously prove Theorem 4.1. In Section C.3, we rigorously prove Theorem 4.2. In Section C.4, we rigorously prove Theorem 4.3. In Section C.5, we rigorously prove Theorem 4.4. In Section C.6, we provide another special case on reducing interval number. In Section D, we provide the detailed experiment settings, results and other analysis.

## A  SOME SUPPLEMENTARY STATEMENTS

### A.1  USE OF LARGE LANGUAGE MODELS (LLMS)

Large language models were used solely as a language-polishing tool to improve grammar, clarity, and style. They did not contribute in any way to the research ideas, theoretical developments, experimental design, data analysis, or interpretation of results. All scientific content was entirely developed by the authors.

### A.2  ETHICS STATEMENT

All authors have read and comply with the ICLR Code of Ethics. Our study uses only publicly available datasets and involves no human subjects, so institutional review board (IRB) approval is not required and no privacy or security risks arise. The research introduces no potentially harmful insights or applications and poses no concerns regarding discrimination, bias, or fairness. It is free of conflicts of interest or sponsorship influences, and all data handling and experimental procedures follow accepted standards of research integrity and legal compliance. Taken together, these considerations ensure that our work satisfies the ethical requirements outlined in the ICLR Code of Ethics for data use, privacy and security, fairness, conflicts of interest, and research integrity.

### A.3  REPRODUCIBILITY STATEMENT

To ensure that our results can be independently verified, we provide a complete account of the SCD-split framework and all experimental settings in the main paper (Sections 3 and 5). The appendix C contains all theoretical assumptions and full proofs of the main theorems, while the appendix D describes further experiment details including data preprocessing, parameter choices, and additional experimental results. The source code is available in the supplementary material. Together, these materials give readers all of the information and resources needed to reproduce both the theoretical and empirical findings of this work.

## B  ADDITIONAL DISCUSSIONS

**Scope of our method: regression.** Regression problems are among the most common settings in practical machine learning applications such as house price prediction, medical risk estimation, and energy demand forecasting, and classical conformal prediction methods—such as CQR (Romano et al., 2019a)—have been primarily developed for regression tasks. The method proposed in this paper is specifically tailored to regression problems, where the prediction set is continuous. In contrast, classification tasks naturally produce discrete prediction sets (i.e., subsets of labels), and thus do not suffer from the same issues of fragmented intervals or interpretability challenges that arise in regression. Therefore, the smoothing technique we introduce is meaningful primarily in the regression setting.

**Why not simply merge nearby intervals?** One might consider a simpler post-processing heuristic, such as merging CD-split intervals that lie within a fixed distance. However, SCD-split offers a fundamental advantage in that it fully preserves the theoretical coverage guarantees of conformal prediction. Any operation that alters prediction sets after their construction based on their geometric

configuration can violate the exchangeability principle between calibration and test samples, thereby invalidating the $1 - \alpha$ coverage guarantee. In contrast, SCD-split integrates the smoothing operation into the conformal procedure before the quantile computation, ensuring that all calibration and test points are treated symmetrically. This principled design provides not only empirical effectiveness but also rigorous theoretical soundness.

**Feasibility of the target number.** One may argue our method cannot work when the user-specified target number $K_{\text{target}}$ exceeds the number of disjoint intervals produced by the original CD-split procedure. However, in practice users typically prefer prediction sets with a relatively small number of disjoint intervals for interpretability and ease of decision making, so $K_{\text{target}}$ is in most cases naturally modest and well below this upper bound. Moreover, if CD-split itself yields fewer disjoint intervals than $K_{\text{target}}$, that outcome is informative: it indicates that the underlying conditional distribution does not support as many distinct high-probability regions as the user initially expected. In such situations, our framework provides a transparent diagnostic and guidance, allowing users to simply adjust $K_{\text{target}}$ downward so that the interpretability constraint aligns with the intrinsic complexity of the data rather than with a prior guess.

**Additional data split for smoothing parameter tuning.** Although reserving a small validation subset to select the smoothing parameter $\sigma$ slightly reduces data efficiency, this data-driven procedure enables us to identify a better parameter in a principled and effective manner, leading to better overall performance. Moreover, such a design is common in conformal prediction; for instance, the RAPS method (Angelopoulos et al., 2022) similarly sets aside a small portion of data to choose its tuning parameter $\tau$.

**Conditional Density Estimation** (CDE). CDE is a challenging problem in modern statistical inference, especially in high-dimensional regimes (Izbicki & Lee, 2017). CDE methods can be grouped into three categories: (a) *parametric methods* assume that $p(y \mid x)$ follows a specific family of distributions (*e.g.*, Gaussian, Exponential) and use maximum likelihood estimation to determine parameters (Bishop, 2006); (b) *non-parametric methods* calculate the conditional density using the ratio of the joint kernel density estimate to the marginal kernel density estimate (Hyndman et al., 1996; De Gooijer & Zerom, 2003; Genius, 2008). Several works based on this method focus on using different methods to tune parameters (Ichimura & Fukuda, 2010; Holmes et al., 2012). Other approaches include different regression methods (Fan et al., 1996; Takeuchi et al., 2009; Izbicki & Lee, 2017) and least-square (Sugiyama et al., 2010); (c) *neural network based methods* combine neural networks with mixture density models called Mixture Density Networks (MDN) (Rothfuss et al., 2019) or combine neural networks with non-parametric methods called Kernel Mixture Networks (KMN) (Ambrogioni et al., 2017). Another promising method of neural networks based CDE is normalizing flow (Trippe & Turner, 2018; Kobyzev et al., 2021).

**Contributions.** Our main contributions are summarized as follows:

- We introduce the number of intervals as a new evaluation metric, complementing interval length, to more comprehensively characterize the interpretability of prediction sets.

- We propose a smoothing-based method in Section 3 that regularizes the estimated conditional density function, reducing unnecessary peaks and ensuring that the number of disjoint intervals in the prediction set is closer to the target number. This improves the interpretability and stability of the resulting prediction sets. Besides, our smoothing approach is general and can be flexibly integrated into any conformal prediction method based on conditional density estimation, including CD-split and HPD-split.

- Theoretical evidence in Section 4 shows that SCD-split is (a) valid, where the empirical coverage is larger than $1 - \alpha$, (b) efficient, where interval length is still acceptable, and (c) connective, where the number of intervals decreases under special structural cases.

- We conduct comprehensive experiments on both synthetic and real-world datasets in Section 5. The results demonstrate that our method achieves a favorable trade-off between interval length and number, leading to the better interpretability among compared methods.

## C OMITTED PROOFS

### C.1 SOME DEFINITIONS AND PROPOSITIONS

**Definition C.1** (Randomized Smoothing). *Given base function $f$ and input $x \in \mathbb{R}^d$, define smoothed function $\tilde{f}$. Specifically,*

$$\tilde{f}^{RS}(x) = \mathbb{E}(f(x + \delta)),$$

*where $\delta \sim \mathcal{N}(0, \sigma^2 I_d)$. We call it $\sigma$-Randomized Smoothing. In practice, we use the Monte Carlo method to deploy $(\sigma, n)$-Randomized Smoothing as,*

$$\tilde{f}_n^{RS}(x) = \frac{1}{n} \sum_{i=1}^{n} f(x + \delta_i), \ \forall x \in \mathbb{R}^d,$$

*where $\delta_1, \delta_2, \cdots, \delta_n \overset{i.i.d.}{\sim} \mathcal{N}(0, \sigma^2 I_d)$.*

**Definition C.2** (Fourier Smoothing with Gaussian low-pass filtering). *Define the Gaussian low-pass filtering function as*

$$H(w) = e^{-2\pi^2 \sigma^2 w^2} = e^{-\alpha w^2},$$

*where the $\alpha$ is bandwidth. Therefore, we can formalize the Fourier smoothing with Gaussian low-pass filtering as*

$$\tilde{f}^{FS}(t) = \frac{1}{2\pi} \int_{\mathbb{R}} \left[ \int_{\mathbb{R}} f(\tau)e^{-iw\tau}d\tau \right] H(w)e^{iwt}dw,$$

*where $\tilde{f}^{FS}$ is smoothed $f$.*

**Definition C.3** (Profile Distance (Izbicki et al., 2021)). *Given $\mathbf{x} \in \mathcal{X}$ and a conditional density estimator $\hat{f}$, we define the estimated conditional CDF*

$$\hat{H}(z \mid \mathbf{x}) := \int_{\{y: \hat{f}(y|\mathbf{x}) \leq z\}} \hat{f}(y \mid \mathbf{x}) \, dy.$$

*The profile distance between $\mathbf{x}_a$ and $\mathbf{x}_b$ is the squared $L^2$ distance between their estimated conditional CDFs:*

$$d^2(\mathbf{x}_a, \mathbf{x}_b) := \int_{-\infty}^{\infty} \left[ \hat{H}(z \mid \mathbf{x}_a) - \hat{H}(z \mid \mathbf{x}_b) \right]^2 dz.$$

**Proposition C.1** (Convergence to the highest predictive density set (Izbicki et al., 2021)). *The highest predictive density set, $\mathcal{C}_{1-\alpha}^*(x)$, is the region with the smallest Lebesgue measure with $1 - \alpha$ coverage:*

$$\mathcal{C}_{1-\alpha}^*(x) := \{y : f(y \mid x) \geq q_\alpha(x)\}, \quad \text{where } q_\alpha(x) \text{ is the } \alpha \text{ quantile of } f(Y \mid x).$$

*A conformal prediction method converges to the highest predictive density set if:*

$$\mathbb{P}\left(Y_{n+1} \in \mathcal{C}_{1-\alpha}^*(X_{n+1})\Delta\mathcal{C}_{1-\alpha}(X_{n+1})\right) = o(1), \quad \text{where } A\Delta B := (A \cap B^c) \cup (B \cap A^c).$$

*The CD-split method satisfies this convergence property when the estimated conditional density $\hat{f}(y \mid x)$ approaches the true density $f(y \mid x)$.*

### C.2 PROOF OF THEOREM 4.1

Throughout we adopt the notation of the main text. Write

$$\mathcal{D}_{n+1} := \{(\mathbf{X}_i, Y_i)\}_{i=1}^{n+1} \quad \text{and} \quad \mathcal{D}_n := \{(\mathbf{X}_i, Y_i)\}_{i=1}^{n}.$$

Assume $\mathcal{D}_{n+1}$ is *exchangeable*. Fix a random split $\mathcal{D}_n = \mathcal{D}_{\text{tr}} \cup \mathcal{D}_{\text{cal}}$ with $|\mathcal{D}_{\text{tr}}| = n_{\text{tr}}$ and $|\mathcal{D}_{\text{cal}}| = m := n - n_{\text{tr}}$. The SCD–split algorithm proceeds in three steps:

(i) **Model fitting.** Using only $\mathcal{D}_{\mathrm{tr}}$ we construct a conditional density estimator $\hat{f}(\,\cdot\mid\mathbf{x})$, then apply Fourier smoothing with a Gaussian kernel to obtain $\tilde{f}^{\mathrm{FS}}(\,\cdot\mid\mathbf{x})$. Both operations are deterministic functions of $\mathcal{D}_{\mathrm{tr}}$; hence $\tilde{f}^{\mathrm{FS}}$ is $\sigma(\mathcal{D}_{\mathrm{tr}})$–measurable.

(ii) **Non-conformity scores.** Define the *density-level score*

$$S\big((\mathbf{x},y);\tilde{f}^{\mathrm{FS}}\big) := \tilde{f}^{\mathrm{FS}}(y\mid\mathbf{x}), \quad (\mathbf{x},y)\in\mathbb{R}^d\times\mathbb{R}.$$

Because $S$ depends on $(\mathbf{x},y)$ only through the symmetric function $\tilde{f}^{\mathrm{FS}}$, it is exchangeable in its arguments conditional on $\mathcal{D}_{\mathrm{tr}}$. For every $(\mathbf{X}_i,Y_i)\in\mathcal{D}_{\mathrm{cal}}$ set $S_i := S\big((\mathbf{X}_i,Y_i);\tilde{f}^{\mathrm{FS}}\big)$ and for the new pair $(\mathbf{X}_{n+1},Y_{n+1})$ set $S_{n+1} := S\big((\mathbf{X}_{n+1},Y_{n+1});\tilde{f}^{\mathrm{FS}}\big)$.

(iii) **Quantile and prediction band.** Let

$$q_\alpha := \mathrm{Quantile}_\alpha\big(S_i\ :\ (\mathbf{X}_i,Y_i)\in\mathcal{D}_{\mathrm{cal}}\big),$$

i.e. the $(\lceil(m+1)(\alpha)\rceil/(m+1))$-th empirical quantile of the calibration scores. The SCD-split prediction band is

$$\mathcal{C}^S_{1-\alpha}(\mathbf{x}) := \big\{y\in\mathbb{R} : S\big((\mathbf{x},y);\tilde{f}^{\mathrm{FS}}\big)\geq q_\alpha\big\}.$$

*Proof. Condition on the training $\sigma$-field* $\mathcal{F}_{\mathrm{tr}} := \sigma(\mathcal{D}_{\mathrm{tr}})$. Given $\mathcal{F}_{\mathrm{tr}}$, the smoothed density $\tilde{f}^{\mathrm{FS}}$ is fixed, while the $(m+1)$ scores

$$\big(S_i : (\mathbf{X}_i,Y_i)\in\mathcal{D}_{\mathrm{cal}}\big) \quad\text{and}\quad S_{n+1}$$

are measurable functions of $\big((\mathbf{X}_i,Y_i)\big)_{i\in\mathcal{I}_{\mathrm{cal}}\cup\{n+1\}}$ and remain *exchangeable* because $\mathcal{D}_{n+1}$ was exchangeable. Standard split-conformal theory (Vovk et al., 2005) then implies

$$\mathbb{P}\big(S_{n+1}\geq q_\alpha\big)\ \geq\ 1-\alpha.$$

Equivalently,

$$\mathbb{P}\big(Y_{n+1}\in\mathcal{C}^S_{1-\alpha}(\mathbf{X}_{n+1})\big)\ \geq\ 1-\alpha.$$

$\qquad\square$

### C.3 PROOF OF THEOREM 4.2

**Definition C.4** (The number of sign variations). *Let $\mathcal{F}$ denote the space of all real-valued measurable and locally bounded functions on $\mathbb{R}$. For any $f\in\mathcal{F}$, we define the number of sign variations of $f$ as*

$$v(f) := \sup\{\mathrm{VarSign}(f(x_1),f(x_2),\ldots,f(x_n))\mid n\in\mathbb{N},\ x_1<x_2<\cdots<x_n\in\mathbb{R}\},$$

*where $\mathrm{VarSign}(a_1,\ldots,a_n)$ is defined to be the number of sign changes in the sequence $(a_1,\ldots,a_n)$ after removing all zero entries. Specifically, let $(a_{i_1},\ldots,a_{i_k})$ be the nonzero subsequence of $(a_1,\ldots,a_n)$, and define $s_j := \mathrm{sgn}(a_{i_j})\in\{-1,+1\}$. Then*

$$\mathrm{VarSign}(a_1,\ldots,a_n) := \sum_{j=1}^{k-1}\mathbb{I}(s_j\cdot s_{j+1}=-1),$$

*where $\mathbb{I}(\cdot)$ denotes the indicator function.*

**Definition C.5** (Variation-Diminishing Transformation). *Let $\Lambda:\mathbb{R}\to\mathbb{R}$, define operator*

$$T[f](x) := \int_{\mathbb{R}}\Lambda(x-t)f(t)dt.$$

*We say $T$ is a **variation-diminishing transformation** if $\forall f:\mathbb{R}\to\mathbb{R}$ and $f$ is bounded and measurable, there holds*

$$v(T[f])\leq v(f).$$

**Lemma C.1** (Theorem in (Schoenberg, 1948)). *For the convolution transformation defined as*

$$g(x) = \int_{\mathbb{R}} \Lambda(x-t) f(t) dt.$$

*It is variation-diminishing if $\Lambda$ and $f$ satisfies*

1. *$0 < \int_{\mathbb{R}} |\Lambda(x)| \, dx < \infty$;*

2. *$f$ is bounded and measurable.*

*That is, if $\Lambda$ and $f$ satisfies conditions above, there holds*

$$v(g) \leq v(f).$$

*Proof.* Without loss of generality, assume $t = 0$, we have

$$M = \left\lfloor \frac{v(f)+1}{2} \right\rfloor,$$

where $M$ is the number of disconnected intervals of $A_t$. Since Lemma C.4 tells us that the fourier smoothing transformation is actually Gaussian Kernel Convolution, the fourier smoothing transformation is variation-diminishing by Lemma C.1. Therefore,

$$M' = \left\lfloor \frac{v(\tilde{f}^{\mathrm{FS}})+1}{2} \right\rfloor \leq \left\lfloor \frac{v(f)+1}{2} \right\rfloor = M.$$

$\square$

## C.4 PROOF OF THEOREM 4.3

*Proof.* Without loss of generality relabel the two components as $(a_1, b_1)$ and $(a_2, b_2)$ with gap $(b_1, a_2)$. Let $m_{bc} := \sup_{x \in (b_1, a_2)} f(x) \leq t - \varepsilon$.

**Step 1: the midpoint rises above the threshold.** Set $x^\star := \frac{b_1 + a_2}{2}$. By convolution,

$$\tilde{f}^{\mathrm{FS}}(x^\star) = \int_{\mathbb{R}} \phi_\sigma(x^\star - s) f(s) \, ds = \int_{(a_1, b_1) \cup (a_2, b_2)} \phi_\sigma(x^\star - s) f(s) \, ds + \int_{b_1}^{a_2} \phi_\sigma(x^\star - s) f(s) \, ds.$$

Since $f(s) \leq m_{bc}$ on the valley, we have

$$\tilde{f}^{\mathrm{FS}}(x^\star) \geq (t - m_{bc}) \int_{|u| \geq \delta/2} \phi_\sigma(u) \, du + m_{bc}.$$

**Step 2: the whole valley rises above the threshold.** The kernel $\phi_\sigma$ is continuous, strictly positive and unimodal, hence $\tilde{f}^{\mathrm{FS}}$ attains its minimum on $[b_1, a_2]$ at the endpoints. But for $s \in (a_1, b_1) \cup (a_2, b_2)$ the integrand contribution to $\tilde{f}^{\mathrm{FS}}(b_1)$ and $\tilde{f}^{\mathrm{FS}}(a_2)$ is no smaller than at $x^\star$, so

$$\tilde{f}^{\mathrm{FS}}(b_1) \geq t, \qquad \tilde{f}^{\mathrm{FS}}(a_2) \geq t.$$

Continuity of $\tilde{f}^{\mathrm{FS}}$ yields $\tilde{f}^{\mathrm{FS}}(x) \geq t$ for every $x \in [b_1, a_2]$. Thus $(a_1, b_2) \subseteq B_t$ and the two components merge.

**Step 3: counting components.** At least one pair of components of $A_t$ has merged, so $M' \leq M - 1$.
$\square$

## C.5 PROOF OF THEOREM 4.4

### C.5.1 TECHNIQUE LEMMA

**Lemma C.2** (Hoeffding's Inequality, Theorem 2.6.2 in (Vershynin, 2018)). *Let $X_1, X_2, \cdots, X_n$ be zero-mean independent sub-Gaussian random variables. Then, for any $t > 0$, we have*

$$\mathbb{P}\left( \left| \sum_{i=1}^{n} X_i \right| \geq t \right) \leq 2 \exp\left( -\frac{ct^2}{\sum_{i=1}^{n} \|X_i\|_{\psi_2}^2} \right),$$

where $c > 0$ is an absolute constant and $\| \cdot \|_{\psi_2}$ is sub-Gaussian norm defined by:

$$\|X\|_{\psi_2} := \inf \left\{ c \geq 0 : \mathbb{E} \left( e^{X^2/c^2} \right) \leq 2 \right\}.$$

**Lemma C.3.** *Assume $f : \mathbb{R} \to \mathbb{R}$ is $L$-Lipschitz. $\forall \eta \in (\frac{1}{2}, 1)$, there exist $C > 0$ an absolute constant, such that with probability $1 - \eta$,*

$$\|f - \tilde{f}_n^{RS}\|_\infty < L\sigma \sqrt{\frac{\log(2/\eta)}{Cn} + \frac{2}{\pi}},$$

*where $\tilde{f}_n^{RS}$ is $(\sigma, n)$-Randomized Smoothed function.*

*Proof.* $\forall x \in \mathbb{R}^d$, there holds

$$|f(x) - \tilde{f}_n^{\text{RS}}(x)| \leq \frac{1}{n} \sum_{i=1}^n |f(x + \delta_i) - f(x)|$$

$$\leq \frac{1}{n} \sum_{i=1}^n L |\delta_i|$$

$$\leq \frac{L}{n} \left| \sum_{i=1}^n (|\delta_i| - \mathbb{E}|\delta_i|) \right| + L\mathbb{E}|\delta_i|.$$

Bound the first term using Lemma C.2. $\forall \eta \in (\frac{1}{2}, 1)$, there exists $C > 0$, such that,

$$\mathbb{P} \left( \frac{L}{n} \left| \sum_{i=1}^n (|\delta_i| - \mathbb{E}|\delta_i|) \right| \geq \sqrt{\frac{L^2\sigma^2 \log(2/\eta)}{Cn}} \right) \leq \eta.$$

Since $\mathbb{E}|\delta_i| = \sigma\sqrt{2/\pi}$ and the upper bound is consistent for all $x \in \mathbb{R}$, we have

$$\|f - \tilde{f}_n^{\text{RS}}\|_\infty < L\sigma \sqrt{\frac{\log(2/\eta)}{Cn} + \frac{2}{\pi}}$$

holds with probability $1 - \eta$. $\qquad\square$

**Lemma C.4.** *Fourier smoothing with Gaussian low-pass filtering is equivalent to the case of randomized smoothing on large samples. Specifically, assume $f$ is $L$-Lipschitz, if $\sigma^2 = 2\alpha$, there holds*

$$\|\tilde{f}_n^{RS} - \tilde{f}^{FS}\|_\infty \xrightarrow{n \to \infty} 0,$$

*where $\sigma^2$ denotes the noise variance of randomized smoothing and $\alpha$ denotes the bandwidth of Fourier smoothing.*

*Proof.* Let

$$K(t - \tau) = \frac{1}{2\pi} \int_\mathbb{R} e^{-\alpha w^2} e^{iw(t-\tau)} dw,$$

we have

$$\tilde{f}^{\text{FS}}(t) = \int_\mathbb{R} f(\tau) K(t - \tau) d\tau.$$

Obviously, it is a kernel function and we convert the Fourier transform and inverse Fourier transform process into a convolution form. We can calculate this kernel function. Let $s = t - \tau$

$$K(s) = \frac{1}{2\pi} \int_\mathbb{R} e^{-\alpha w^2} e^{iws} ds$$

$$= \frac{1}{2\pi} \int_\mathbb{R} e^{-\alpha w^2 + iws} ds$$

$$= \frac{1}{2\sqrt{\pi\alpha}} \exp \left\{ -\frac{s^2}{4\alpha} \right\}.$$

Therefore, this kernel function is a Gaussian kernel with $\sigma^2 = 2\alpha$. Then let's investigate randomized smoothing.

$$\tilde{f}^{\mathrm{RS}} = \int_{\mathbb{R}} f(x + \delta)p(\delta)d\delta$$

$$\overset{t=x+\delta}{=\!=\!=} \int_{\mathbb{R}} f(t)p(t - x)dt$$

$$= \int_{\mathbb{R}} f(t)p(x - t)dt,$$

where $p(x)$ is the density function of the Gaussian distribution with $\sigma^2$ as the variance. Therefore, we have

$$\|\tilde{f}_n^{\mathrm{RS}} - \tilde{f}^{\mathrm{FS}}\|_\infty \leq \|\tilde{f}_n^{\mathrm{RS}} - \tilde{f}^{\mathrm{RS}}\|_\infty + \|\tilde{f}^{\mathrm{RS}} - \tilde{f}^{\mathrm{FS}}\|_\infty = \|\tilde{f}_n^{\mathrm{RS}} - \tilde{f}^{\mathrm{RS}}\|_\infty.$$

Let $Z_n(x) = \tilde{f}_n^{\mathrm{RS}}(x) - \tilde{f}^{\mathrm{RS}}(x) = \int_{\mathbb{R}} f(x + t)(\mu_n - \mu)dt$, where $\mu_n = \frac{1}{n}\sum_{i=1}^n \delta_{\delta_i}$ is an empirical measure and $\mu$ denotes the Gaussian probabilistic measure, $\forall x \in \mathbb{R}$, we have

$$Z_n(x) = \int_{\mathbb{R}} f(x + t)(\mu_n - \mu)(dt)$$

$$= \int_{\mathbb{R}} (f(x + t) - f(x))(\mu_n - \mu)(dt).$$

Therefore,

$$|Z_n(x)| \leq \int_{\mathbb{R}} |f(x + t) - f(x)|(\mu_n - \mu)(dt)$$

$$\leq \int_{\mathbb{R}} L|t|(\mu_n - \mu)(dt)$$

$$= \frac{1}{n}\sum_{i=1}^n |\delta_i| - \mathbb{E}|\delta_i|.$$

Since the upper bound is consistent w.r.t. $x$, by law of large number, we have

$$\|\tilde{f}_n^{\mathrm{RS}} - \tilde{f}^{\mathrm{FS}}\|_\infty \leq \|\tilde{f}_n^{\mathrm{RS}} - \tilde{f}^{\mathrm{RS}}\|_\infty = \|Z_n\|_\infty \overset{n\to\infty}{\longrightarrow} 0.$$

$\square$

### C.5.2 Full Proof of Theorem 4.4

*Proof.* For simplicity, we assume the density function is unimodal at first. Let $\{i_1, \cdots, i_{n_j}\} = \{i : X_i \in A(x_{n+1})\}$, $U_l = \widehat{f}(y_{i_l} \mid x_{i_l})$, for $l = 1, \cdots, n_j$, and $U_{n_j+1} = \widehat{f}(y_{n+1} \mid x_{n+1})$. Similarly, let $\tilde{U}_k = \tilde{f}^{\mathrm{FS}}(y_{i_k} \mid x_{i_k})$, for $k = 1, \cdots, n_j$, and $\tilde{U}_{n_j+1} = \tilde{f}^{\mathrm{FS}}(y_{n+1} \mid x_{n+1})$. By Lemma C.3 and Lemma C.4, we have with probability $1 - \eta$,

$$|U_i - \tilde{U}_i| < L\sigma\sqrt{\frac{2}{\pi}}, \quad \forall i \in 1, \cdots, n_j + 1.$$

Let $t$ be the oracle original quantile threshold and $\varepsilon = L\sigma\sqrt{2/\pi}$, we have

$$\mathbb{P}\left(\tilde{U}_{n+1} \geq t - \varepsilon \mid x_{n+1}\right) \overset{(i)}{\geq} \mathbb{P}\left(U_{n+1} \geq t \mid x_{n+1}\right)$$

$$\overset{(ii)}{=} \mathbb{P}\left(U_i \geq t \mid x_i\right)$$

$$\geq 1 - \alpha,$$

where (i) holds since the error control of $U_l$ and (ii) follows from the definition of the profile of the density. Therefore, let $g = f + \varepsilon$. Since the density function is unimodel, $f(y_{n+1} \mid x_{n+1}) = t$ only has two solution. Denote $[\ell, u] = \{y : f(y \mid x_{n+1}) \geq t\}$ and $[\tilde{\ell}, \tilde{u}] = \{y : g(y \mid x_{n+1}) \geq t - \varepsilon\}$, there holds

$$|\tilde{l} - l| \leq \left||\tilde{\ell} - \tilde{u}| - |\ell - u|\right| \leq \left|\tilde{\ell} - \ell\right| + |\tilde{u} - u| \overset{(i)}{\leq} 4\frac{|f - g|}{M} = \frac{4\varepsilon}{M} = \frac{4L\sigma}{M}\sqrt{\frac{2}{\pi}},$$

where (i) holds since $\left| \widehat{f}(y_1 \mid x) - \widehat{f}(y_2 \mid x) \right| \geq M \left| y_1 - y_2 \right|$. Since the number of disconnected intervals returned by CD-split is $N$ and the result in Theorem 4.2 demonstrates that the number of disconnected intervals doesn't increase, the result of the multimodel distribution is as follows

$$|\tilde{l} - l| \leq \frac{4NL\sigma}{M}\sqrt{\frac{2}{\pi}}. \tag{10}$$

$\square$

### C.6 ANOTHER SPECIAL CASE ON REDUCING INTERVAL NUMBER: HIGH–FREQUENCY SMALL–AMPLITUDE PERTURBATIONS

The following result shows that when the disconnected components of $A_t$ are created *solely* by a high–frequency oscillatory perturbation, Gaussian-kernel Fourier smoothing suppresses those oscillations and thus strictly reduces the component count.

**Definition C.6** (($\sigma, t$)–HF perturbation)**.** *Let $g : \mathbb{R} \to \mathbb{R}$ be bounded, measurable and differentiable and fix $t \in \mathcal{T}$. Set the safety gap*

$$\Delta(t, g) := \inf_{x \in \mathbb{R}} \left| g(x) - t \right| \in (0, \infty). \tag{\dagger}$$

*For parameters $\varepsilon > 0$, $k > 0$, $\sigma > 0$ we call*

$$f(x) := g(x) + \varepsilon \sin(kx)$$

*an $(\sigma, t)$–high-frequency perturbation of $g$ if*

(i) $\Delta(t, g) < \varepsilon$   *(the oscillation amplitude is large enough to cross the threshold);*

(ii) *the attenuated amplitude after Gaussian convolution,*

$$\varepsilon_\sigma := \varepsilon\, e^{-2\pi^2 \sigma^2 k^2},$$

*satisfies $\varepsilon_\sigma < \Delta(t, g)$   (the residual amplitude is too small to cross $t$).*

**Theorem C.1** (Strict reduction for an HF perturbation)**.** *Adopt the setting of Definition C.6 and write $A_t(f) = \bigsqcup_{j=1}^{M}(a_j, b_j)$ and $B_t(f) = \bigsqcup_{j=1}^{M'}(a'_j, b'_j)$ for $f(x) = g(x) + \varepsilon \sin(kx)$ and its Fourier-smoothed version $\tilde{f}^{\mathrm{FS}} = T_\sigma[f] = \phi_\sigma * f$, respectively. Then*

$$M' = M\big(v(g)\big) \quad and \quad M' < M,$$

*hence the number of disconnected intervals strictly decreases.*

*Proof.* **Step 1:** Because $\Delta(t, g) > 0$, the function $g$ stays uniformly away from the threshold, so $g(x) - t$ keeps a fixed sign. Consequently $v(g) = 0$ and $M\big(v(g)\big) = \lfloor (0+1)/2 \rfloor = 0$ or $1$. Denote this value by $M_g$.

**Step 2: creation of extra components before smoothing.** Since $\varepsilon > \Delta(t, g)$, the oscillatory term produces at least two distinct roots of $f(x) - t$ within every interval of length $2\pi/k$ where $|g(x) - t| < \varepsilon$. Hence $v(f) \geq 2$ and $M \geq \lfloor (2+1)/2 \rfloor = 1 + M_g$; in particular $M > M_g$.

**Step 3: destruction of the extra components after smoothing.** Because $\tilde{f}^{\mathrm{FS}}(x) = g(x) + \varepsilon_\sigma \sin(kx)$ and $\varepsilon_\sigma < \Delta(t, g)$, we have $\mathrm{sgn}\big(\tilde{f}^{\mathrm{FS}}(x) - t\big) = \mathrm{sgn}\big(g(x) - t\big)$ for all $x$. Thus $v(\tilde{f}^{\mathrm{FS}}) = v(g)$ and $M' = \lfloor (v(g) + 1)/2 \rfloor = M_g$.

**Step 4: comparison.** Combining Steps 2 and 3 gives $M' < M$, completing the proof. $\square$

## D EXPERIMENT DETAILS

### D.1 EXPERIMENT SETTINGS AND RESULTS

**Synthetic data.** First we introduce the simple case. The covariate vector $X = (X_1, \ldots, X_5)$ is sampled *i.i.d.* from $\mathrm{Unif}(-5, 5)$ and standardized, and the response variable Y given X follows

$$Y \mid X \sim \frac{1}{3}\mathcal{N}(0 + 0.1X_1, 0.2^2) + \frac{1}{3}\mathcal{N}(1.0 + 0.1X_1, 0.2^2) + \frac{1}{3}\mathcal{N}(2.0 + 0.1X_1, 0.2^2).$$

Table 2: Results on Synthetic Datasets

| Method | simple | | | complex | | |
|--------|--------|------|------|---------|------|------|
| | Cov. | Len. | Num. | Cov. | Len. | Num. |
| Vanilla CP | $90.32 \pm 0.95$ | $2.45 \pm 0.03$ | $1.00 \pm 0.00$ | $89.98 \pm 0.59$ | $28.70 \pm 0.69$ | $1.00 \pm 0.00$ |
| CQR | $90.28 \pm 0.76$ | $2.44 \pm 0.02$ | $1.00 \pm 0.00$ | $90.36 \pm 0.50$ | $27.48 \pm 1.55$ | $1.00 \pm 0.00$ |
| Local CP | $89.79 \pm 0.89$ | $2.46 \pm 0.03$ | $1.00 \pm 0.00$ | $90.14 \pm 0.69$ | $27.06 \pm 1.17$ | $1.00 \pm 0.00$ |
| DIST | $89.60 \pm 0.91$ | $2.45 \pm 0.02$ | $1.00 \pm 0.00$ | $90.45 \pm 2.64$ | $23.20 \pm 3.99$ | $1.00 \pm 0.00$ |
| CD-split | $89.52 \pm 1.02$ | $\mathbf{2.23} \pm 0.05$ | $2.60 \pm 0.12$ | $91.06 \pm 3.55$ | $21.76 \pm 6.74$ | $2.85 \pm 1.18$ |
| HPD-split | $89.93 \pm 0.85$ | $2.25 \pm 0.04$ | $2.71 \pm 0.71$ | $92.66 \pm 4.89$ | $23.32 \pm 8.86$ | $3.04 \pm 1.60$ |
| **SCD-split** | $89.09 \pm 0.90$ | $2.37 \pm 0.04$ | $\mathbf{2.00} \pm 0.06$ | $89.39 \pm 0.85$ | $\mathbf{16.11} \pm 0.82$ | $\mathbf{1.99} \pm 0.10$ |

Table 3: Results on Real-world Datasets

| Method | bio | | | bike | | |
|--------|-----|------|------|------|------|------|
| | Cov. | Len. | Num. | Cov. | Len. | Num. |
| Vanilla CP | $90.33 \pm 0.85$ | $2.12 \pm 0.12$ | $1.00 \pm 0.00$ | $90.20 \pm 1.11$ | $1.23 \pm 0.14$ | $1.00 \pm 0.00$ |
| CQR | $89.70 \pm 0.91$ | $1.64 \pm 0.11$ | $1.00 \pm 0.00$ | $89.87 \pm 0.92$ | $0.98 \pm 0.09$ | $1.00 \pm 0.00$ |
| Local CP | $90.06 \pm 0.67$ | $1.90 \pm 0.11$ | $1.00 \pm 0.00$ | $89.79 \pm 0.95$ | $1.01 \pm 0.08$ | $1.00 \pm 0.00$ |
| DIST | $90.23 \pm 2.17$ | $1.90 \pm 0.22$ | $1.00 \pm 0.00$ | $89.28 \pm 1.41$ | $0.31 \pm 0.01$ | $1.00 \pm 0.00$ |
| CD-split | $96.59 \pm 2.39$ | $2.29 \pm 0.25$ | $1.36 \pm 0.55$ | $86.76 \pm 1.11$ | $0.22 \pm 0.01$ | $1.07 \pm 0.01$ |
| HPD-split | $98.76 \pm 3.37$ | $2.61 \pm 0.29$ | $1.37 \pm 1.19$ | $89.18 \pm 0.84$ | $\mathbf{0.20} \pm 0.01$ | $1.11 \pm 0.02$ |
| **SCD-split** | $89.23 \pm 0.95$ | $\mathbf{1.52} \pm 0.06$ | $\mathbf{1.49} \pm 0.05$ | $89.00 \pm 1.34$ | $0.25 \pm 0.01$ | $\mathbf{1.01} \pm 0.01$ |

Second, we introduce the complex case. The covariate vector $X = (X_1, \ldots, X_5)$ is sampled *i.i.d.* from $\mathcal{N}(0, 1)$ and standardized, and the response variable Y given X follows

$$Y \mid X \sim \sum_{k=1}^{K} \frac{\exp(X^\top \beta_k)}{\sum_{j=1}^{K} \exp(X^\top \beta_j)} \mathcal{N}(\mu_{\text{base},k} + X^\top \gamma_k, \sigma_k^2),$$

where K = 7 is the number of Gaussian mixture components, and the base means and standard deviations are set as

$$\mu_{\text{base}} = (-15, -10, -5, 0, 5, 10, 15), \quad \sigma = (1, 1.2, 1.5, 1, 1.5, 1.2, 1).$$

The coefficient matrices $\beta_k \in \mathbb{R}^d$ and $\gamma_k \in \mathbb{R}^d$ are randomly initialized with entries drawn from $\mathcal{N}(0, 1)$ and $\mathcal{N}(0, 0.5^2)$, respectively, and they control the conditional mixture weights and component-wise shifts in the conditional means.

**Real-world data.** We evaluate our method on two widely used real-world datasets. The `Bike Sharing` dataset contains 10,886 samples and 18 variables. The `Bio` dataset comprises 45,730 samples and 9 variables.

**Experimental setup.** For all experiments, we randomly draw 2,000 samples for conformal prediction (equally divided between training and calibration sets) and 5,000 samples for testing. All features are standardized before model fitting. We use random forest as the base model for conditional density estimation in all the experiments. Each experiment is repeated across 10 independent trials to ensure statistical reliability. All experiments are conducted using standard CPU environments without the need for GPUs, with modest runtime requirements well within a practical and reproducible range.

Table 4: Different smoothing techniques on synthetic complex dataset

| Method | Coverage (%) | Length | Number of Intervals |
|---|---|---|---|
| Original | $91.06 \pm 3.55$ | $21.76 \pm 6.74$ | $2.85 \pm 1.18$ |
| Fourier | $89.23 \pm 0.77$ | $\mathbf{16.11} \pm 0.68$ | $1.99 \pm 0.01$ |
| Gaussian kernel | $89.40 \pm 0.87$ | $16.90 \pm 1.56$ | $1.95 \pm 0.05$ |
| Spline | $89.30 \pm 0.78$ | $16.37 \pm 0.81$ | $1.95 \pm 0.12$ |
| LOESS | $89.46 \pm 0.80$ | $16.95 \pm 2.11$ | $1.98 \pm 0.01$ |

Table 5: Different loss function on synthetic complex dataset

| Method / $\sigma$ | Global L1 | Global L2 | MAE | MSE |
|---|---|---|---|---|
| CD-split ($\sigma = 0$) | $0.80 \pm 0.29$ | $0.93 \pm 1.08$ | $1.12 \pm 0.14$ | $2.06 \pm 2.02$ |
| SCD-split ($\sigma = 1$) | $0.38 \pm 0.06$ | $0.21 \pm 0.04$ | $0.77 \pm 0.03$ | $1.09 \pm 0.17$ |
| SCD-split ($\sigma = 1.5$) | $\mathbf{0.15} \pm 0.03$ | $\mathbf{0.05} \pm 0.01$ | $0.59 \pm 0.00$ | $0.66 \pm 0.00$ |
| SCD-split ($\sigma = 2$) | $0.33 \pm 0.02$ | $0.13 \pm 0.01$ | $\mathbf{0.56} \pm 0.00$ | $\mathbf{0.57} \pm 0.00$ |
| SCD-split ($\sigma = 5$) | $0.81 \pm 0.00$ | $0.67 \pm 0.00$ | $0.81 \pm 0.00$ | $0.81 \pm 0.00$ |
| SCD-split ($\sigma = 10$) | $1.00 \pm 0.00$ | $1.00 \pm 0.00$ | $1.00 \pm 0.00$ | $1.00 \pm 0.00$ |

### D.2    OTHER ANALYSIS ON EXPERIMENTS

**Choice of loss for validation.** In Table 5, we evaluate performance on the validation set using different loss function for each candidate smoothing parameter $\sigma$. Here we consider four loss functions:

$$R_{\text{Global-L1}}(\sigma) = \left| \frac{1}{|\mathcal{D}_{\text{val}}|} \sum_{(\mathbf{X}_j, Y_j) \in \mathcal{D}_{\text{val}}} N_\sigma(\mathbf{X}_j) - K_{\text{target}} \right|,$$

$$R_{\text{Global-L2}}(\sigma) = \left( \frac{1}{|\mathcal{D}_{\text{val}}|} \sum_{(\mathbf{X}_j, Y_j) \in \mathcal{D}_{\text{val}}} N_\sigma(\mathbf{X}_j) - K_{\text{target}} \right)^2,$$

$$R_{\text{MAE}}(\sigma) = \frac{1}{|\mathcal{D}_{\text{val}}|} \sum_{(\mathbf{X}_j, Y_j) \in \mathcal{D}_{\text{val}}} \left| N_\sigma(\mathbf{X}_j) - K_{\text{target}} \right|,$$

$$R_{\text{MSE}}(\sigma) = \frac{1}{|\mathcal{D}_{\text{val}}|} \sum_{(\mathbf{X}_j, Y_j) \in \mathcal{D}_{\text{val}}} \left( N_\sigma(\mathbf{X}_j) - K_{\text{target}} \right)^2.$$

For $L_1$-type and $L_2$-type losses, the table reports these losses as functions of $\sigma$ and demonstrates that they exhibit the same trend across all candidates, leading to the same choice of $\hat{\sigma}$. Hence, the selection of the smoothing parameter is insensitive to whether $L_1$ or $L_2$ is used. In our main algorithm, we therefore adopt the $L_1$ loss for validation. *Moreover*, the distinction between *Global* (outer) and *Inner* (MAE/MSE) aggregation reflects different goals: the global losses aim to match the target number of intervals in an average sense, whereas the inner losses measure how close each individual prediction set is to the target. Therefore, one can also adopt MAE or MSE and correspondingly tune the smoothing parameter if the goal is to make the number of intervals for every single test point as close as possible to $K_{\text{target}}$.

**Stability Across Trials.** We further evaluate the robustness of each method by examining the variability of performance across multiple random trials. Specifically, we measure the standard deviation of coverage, interval length, and interval number across different runs. Our results show that after applying smoothing techniques, the standard deviations of all three metrics are consistently reduced across both synthetic and real-world datasets. This indicates that smoothing not only improves the interpretability and efficiency of the prediction sets, but also enhances the stability of the predictions, making the results more reliable under different random splits or sampling variations.

