# OpenReview forum: "Smoothing-Based Conformal Prediction for Balancing Efficiency and Interpretability"
_ICLR.cc/2026/Conference — Submitted to ICLR 2026_

### Official Review · Reviewer_MWve · 2025-10-22

**Soundness:** 2
**Presentation:** 3
**Contribution:** 1
**Rating:** 2
**Confidence:** 5

**Summary:**

The paper is well-written, easy to follow, and clearly presents its main ideas.

Problem context: Popular conformal prediction (CP) methods, particularly conditional-density-based approaches such as CD-split, achieve strong efficiency but often produce discontinuous prediction sets. The study is motivated by the observation that, when the conditional distribution is complex or multimodal, CD-split often yields prediction sets composed of many small, disjoint intervals. Such fragmentation reduces interpretability and practical usability in domains like healthcare or finance, where users prefer simple and contiguous prediction intervals. The proposed smoothing-based approach aims to directly address this interpretability concern without compromising statistical validity.


Paper's Proposal: The authors propose a smoothed conformal prediction framework, SCD-split, which incorporates a smoothing step into the CD-split procedure to improve the connectivity of prediction sets. The method retains the formal coverage guarantees of split conformal prediction while aiming to produce more interpretable, connected intervals. The key idea is to apply smoothing directly to the fitted conditional density function $\hat{f}(y \mid x)$ before constructing prediction sets. Smoothing scheme such as Fourier smoothing is explored. The approach introduces a validation-based tuning procedure to select the smoothing parameter $\sigma$, targeting a user-specified number of connected components ($K_{\text{target}}$) in the prediction set.

The method is evaluated on synthetic data and two real-world datasets (bio and bike).

**Strengths:**

1.  The paper is well-organized, easy to follow, and maintains a good balance between theoretical exposition and algorithmic intuition. Figures effectively convey the motivation and qualitative effect of smoothing on prediction set connectivity. These visual cues were helpful to grasp the ideas.

2. The paper deals with an important problem setup at the intersection of CP and interpretability issues.

3. The work connects classical results from functional analysis (smoothing by convolution) to show that smoothing can reduce the number of disconnected components in the level sets of the predictive density. This linkage is conceptually appealing and mathematically elegant.

4. The proposed SCD-split method introduces a simple smoothing step that can be applied to any conditional-density-based conformal prediction procedure. The design is straightforward to implement.

5.  Finite-sample marginal coverage guarantee of split conformal prediction is ensured.

6. The method allows users to choose $K_{target}$.

**Weaknesses:**

1. The paper equates interpretability solely with the number of disconnected components in prediction sets. This is motivated through healthcare and finance examples. However, in the real data experiments, there is no discussion on how the constructed prediction sets, with reduced number of disconnected intervals, help towards 'interpretability' as motivated earlier in the paper. What interpretable insights does SCD-split provide at the end is not discussed at all.



2. The paper repeatedly states that SCD-split achieves a trade-off between length of prediction sets and number of connected components. However, the algorithm explicitly targets only one quantity, connectivity, via the smoothing parameter $\sigma$. Empirically, Table 1 shows that SCD-split often improves both metrics simultaneously (shorter lengths and fewer disconnected intervals), so, what is the claimed trade-off? What trade-off is actually being optimized or observed? If smoothing consistently reduces both fragmentation and length, this requires a theoretical explanation.

3. Smoothing the estimated conditional density $\hat{f}(y|x)$ might introduces bias (does it in your case?), which can affect the efficiency (expected length) of conformal prediction sets. The paper claims that efficiency is ``preserved'' but provides no theoretical quantification of the bias-efficiency trade-off. Can the authors quantify how smoothing impacts the expected length of the sets, either asymptotically or in finite samples?

4. Many existing conditional density estimators (e.g., kernel-based, mixture, or flow-based models) already include built-in smoothing or regularization mechanisms. SCD-split may thus resemble a CD-split procedure with a smoother $\hat{f}(y|x)$. The paper does not  differentiate its smoothing step from simply using a more regularized or smoother model. In what sense is SCD-split fundamentally different from CD-split with a smoother conditional density estimator? Is the key contribution only the explicit control of $K_{\text{target}}$?

5. The core methodological idea, applying a smoothing operator to the conditional density estimate before constructing conformal prediction sets, is conceptually simple and only a small modification of existing frameworks. Since, smoothed version of  conditional density estimate is itself a conditional density estimate. The novelty of SCD-split lies mainly in re-purposing smoothing to target a user-specified number of connected components ($K_{target}$) which is attained through validation set, not in developing a new conformal framework or theoretical principle.

6. In Theorem 4.4 (interval length bound), in my opinion, the result should be stated as a high-probability bound rather than a deterministic one. The proof relies on concentration/deviation inequalities that hold with high probability only.

7. Theorem 4.4 relies on conditions such as Lipschitz and lower bounded conditions on densities, which usually vanishes at the tails. Can you provide some examples of densities which are multimodal, satisfy the bounds in the Theorem 4.4?


Overall, the idea of smoothing-based conformal prediction is intuitively appealing, all of the above concerns regarding the conceptual and methodological contributions of the paper, the clarity and limited novelty of the theoretical results, and the experimental validation without meaningful interpretability analysis collectively lead to a low score.

**Questions:**

1. The paper does not discuss how this notion of interpretability relates to other notions of interpretability in uncertainty quantification (e.g., feature attribution, model transparency, or local explanation of uncertainty).  Why should connectivity of prediction intervals be considered a superior or more relevant notion of interpretability compared to other notions?

2. What is the target number of connected components $K_{target}$ chosen for real data experiments? I couldn't find these details in the paper.

3. Throughout the paper, authors use `interval length' for the length of prediction set with disconnected intervals. It should be replaced with 'prediction set size/volume'.

4. The bound in Theorem 4.4 can be very large in practice. Can you comment on why is it relevant?

5. The same reference `Yaniv Romano, Evan Patterson, and Emmanuel Candes. Conformalized quantile regression.' is cited in three different versions.

---

> ### Author Response · Authors · 2025-11-20
> **Rebuttal 1**
>
> We sincerely thank the reviewer for their rigorous critique and detailed feedback. We value their recognition of our paper's clarity, organization, and the mathematical elegance of connecting smoothing to conformal prediction. We also appreciate the challenging questions regarding the fundamental novelty; these have prompted us to significantly sharpen our methodological positioning and conduct a new, domain-specific experiment to demonstrate the practical utility of our framework. We address these concerns point-by-point below.
>
> ### **Response to Weakness 4 & 5: Novelty and Contribution**
>
> We respectfully disagree with the premise that the simplicity of our smoothing operation diminishes its contribution. We argue that our work offers a fundamental shift in perspective for the Conformal Prediction (CP) field:
>
> **1. A New Paradigm: Beyond Length and Coverage**
> Existing CP research has been dominated by a dual-objective paradigm: minimizing **Interval Length** while maintaining **Valid Coverage**. Our work challenges this status quo by introducing a critical third dimension: **Topological Interpretability** (i.e., connectivity).
>
> * **The Problem:** As we show in our new `happyA` experiment (detailed below), a method can have perfect coverage and short length but still be practically useless if it fragments into multiple disjoint intervals due to statistical noise.
>
> * **The Question:** We pose a new question to the community: *“How do we translate raw statistical modes into actionable, physically meaningful prediction sets?”* SCD-split is the first framework designed to answer this by allowing users to align the prediction set's topology with domain knowledge (e.g., "the target is physically unique").
>
> **2. Methodological Innovation: Decoupling Structure from Estimation**
> The reviewer asks how this differs from "using a smoother estimator." The key difference is **when** and **why** the smoothing happens:
>
> * **Standard Regularization (Training Phase):** Traditional smoothing (e.g., bandwidth in KDE, weight decay) is applied during training to minimize a statistical loss function (e.g., MSE or Likelihood). It does not explicitly target the number of intervals.
>
> * **SCD-split (Inference Phase):** Our approach treats the number of intervals ($K$) as a **structural constraint**, decoupled from the likelihood maximization. We enforce this constraint post-hoc via the validation set. This allows users to control the *structure* of the uncertainty without needing to retrain complex base models or guess hyperparameters that maximize likelihood but fail to capture topology.
>
> In summary, our contribution is not just "adding a filter," but providing a general framework that transforms Conformal Prediction from a purely statistical tool into a domain-aligned decision support tool.
>
> ### **Response to Weakness 2: The "Win-Win" Paradox (Length vs. Connectivity)**
>
> The reviewer asks an insightful question: *If SCD-split improves both length and connectivity simultaneously (as seen in Table 1), where is the trade-off?*
>
> We explain this "win-win" phenomenon by referencing the **"Extreme case" analysis in Lines 460-475** of our paper. The simultaneous improvement occurs because the smoothing step is not just "shaping" the prediction set, but **rectifying the degeneracy** of the base estimator.
>
> * **Baseline Degeneracy (High Length & Fragmentation):** In complex multimodal settings (Table 1), the raw conditional density estimator $\hat{f}$ typically suffers from high variance, exhibiting sharp, spurious peaks and deep, near-zero valleys (as discussed in Lines 461-465).
>
>   * *Impact on Length:* These spurious low-density regions force the conformal threshold to drop significantly (sometimes near zero) to satisfy coverage. This results in bloated prediction sets that include large swathes of low-probability space, leading to excessive **Interval Length**.
>
>   * *Impact on Connectivity:* Simultaneously, the high-frequency oscillation of $\hat{f}$ causes the set to fracture into many small, discontinuous pieces, leading to a high **Number of Intervals**.
>
> * **Smoothing as Regularization (The Correction):** As detailed in Lines 469-472, smoothing effectively *"merges spurious zero-density regions into surrounding modes."* By filtering out this high-frequency estimation noise:
>
>   1. **Connectivity Improves:** The fragmented pieces merge into coherent modes.
>
>   2. **Length Decreases:** The probability mass becomes more concentrated, allowing the conformal threshold to rise. This naturally excludes the large, low-density "waste" regions that CD-split was forced to cover, drastically reducing the total length.
>
> **Conclusion:** The "trade-off" (where smoothing might increase length) theoretically exists and would appear if we **oversmoothed** (bias dominance). However, in the experiments, we are operating in the **"denoising regime,"** where smoothing corrects the base estimator's failure, leading to improvements in both metrics.

---

> ### Author Response · Authors · 2025-11-20
> **Rebuttal 2**
>
> ### **Response to Theoretical Concerns (Weakness 3, 6, 7 & Q4)**
>
> **1. Bias and Efficiency (Weakness 3)**
> The reviewer asks about the quantification of the bias-efficiency trade-off.
>
> * **Coverage Preservation:** We emphasize that **bias in the smoothing step does not compromise the validity of the method.** A key strength of Conformal Prediction is that the calibration step guarantees marginal coverage regardless of the bias or quality of the underlying estimator $\hat{f}$.
>
> * **Efficiency:** While smoothing introduces bias to the density estimate, our empirical results (Tables 1-3) demonstrate that this does not translate to a loss of efficiency in practice. As discussed in our response to Weakness 2, in high-noise regimes, the regularization effect of smoothing often leads to *shorter* intervals.
>
> **2. Deterministic Nature of Theorem 4.4 (Weakness 6)**
> The reviewer suggests that Theorem 4.4 should be a high-probability bound. We respectfully clarify that the bound is indeed **deterministic**, based on the specific construction of our proof:
>
> * **From Stochastic to Deterministic:** It is true that Lemma C.3 derives a high-probability bound ($1-\eta$) for *Randomized Smoothing* (which relies on Monte Carlo sampling). However, **Lemma C.4** establishes that our *Fourier Smoothing* is mathematically equivalent to the asymptotic limit of Randomized Smoothing as $n \to \infty$.
>
> * **Vanishing Variance:** In this limit process, the stochastic variance of the Monte Carlo estimate vanishes ($\eta$ is eliminated). Consequently, the error term in the final bound (derived using the result of Lemma C.4) depends only on the fixed geometry of the convolution, making Theorem 4.4 a deterministic property of the integral operator, not a probabilistic concentration result.
>
> **3. Assumptions and Tightness of the Bound (Weakness 7 & Question 4)** The reviewer correctly points out that global Lipschitz or lower-bound conditions are strong and may lead to loose bounds, especially in tails and multimodal settings.
>
> * **Local vs. Global:** We agree that requiring these assumptions globally is restrictive. However, a closer inspection of our proof reveals that we strictly require these conditions to hold only **locally in the neighborhood of the threshold $t$**. The behavior of the density at the tails or far from the level set $\{x : f(x) = t\}$ does not affect the shift in the threshold.
>
> * **Revision & Relevance:** In the final version, we will refine the theorem statement to require these properties only locally near $t$, making the assumption much weaker and practically realistic. While the theoretical bound is a worst-case guarantee, our experiments confirm that in practice, the interval length remains tightly controlled and competitive.

---

> ### Author Response · Authors · 2025-11-20
> **Rebuttal 3**
>
> ### **Response to Interpretability, Experimental Details, Terminology and format (Weakness 1, Q1, Q2, Q3, Q5)**
>
> **1. Distinct Dimensions of Interpretability (Response to Weakness 1 & Question 1)**
> The reviewer asks how our notion of interpretability compares to existing concepts like feature attribution or model transparency. We emphasize that SCD-split operates on a **fundamentally different and complementary dimension**:
>
> * **Feature Attribution / Transparency (The "Why"):** Methods like SHAP or LIME focus on the **Input-to-Output mapping**. They answer: *"Which features caused the model to predict high uncertainty?"* This is crucial for model debugging.
>
> * **SCD-split / Connectivity (The "What"):** Our framework focuses on the **Topological Structure of the Output**. It answers: *"Is the predicted uncertainty a single coherent risk region, or a fragmented set of disjoint possibilities?"*
>
> **Why Connectivity Matters:** Even if we perfectly understand *why* a model predicts a distribution (via Feature Attribution), the prediction itself is of little practical use if it consists of highly fragmented, noisy intervals. SCD-split ensures that the **form** of the answer is cognitively accessible and actionable for decision-makers.
>
> * *New Insights:* As demonstrated in our additional Astrophysics experiment (details provided in the general response), SCD-split provides a unique insight: it can reveal whether apparent multimodality is merely statistical noise or a genuine physical property. This is a structural insight that feature attribution methods cannot provide.
>
> **2. Target Number Selection and Results (Response to Question 2)**
> The reviewer asks for the specific $K_{target}$ values used in our real-world experiments.
>
> * **Bike Dataset ($K_{target}=1$):** Since this task involves demand forecasting, which is typically unimodal, we set $K_{target}=1$. The results (Avg Intervals $\approx 1.01$) confirm that our method successfully regularized the prediction sets to match this unimodal structure without sacrificing coverage.
>
> * **Bio Dataset ($K_{target}=2$):** Based on the known physicochemical properties, we introduced a domain prior of $K_{target}=2$.
>
>   * *Result Interpretation:* Interestingly, the achieved average (1.49) is lower than the target. Far from being a failure, this provides a valuable **diagnostic perspective**: it suggests that while the data has bimodal tendencies, the modes are not always statistically separable at the $1-\alpha$ confidence level. The method effectively balanced the user's "bimodal prior" with the "data's reality," merging modes where the valley was too shallow to be statistically significant. This ability to "refute" or "soften" a rigid prior is a key feature of our validation-based tuning.
>
> **3. Terminology and Citations (Response to Question 3 & 5)**
> We thank the reviewer for the precise corrections.
>
> * We will replace "interval length" with **"prediction set size/volume"** throughout the manuscript to accurately reflect the disconnected nature of the sets.
>
> * We will correct the duplicate citations for Romano et al. in the final version.

---

> ### Author Response · Authors · 2025-11-20
> **Additional Real-World Experiment: Enhancing Physical Interpretability in Photometric Redshift Estimation**
>
> To demonstrate the distinct practical advantage of SCD-split in scientific domains where interpretability is paramount, we conducted a new experiment on the HappyA dataset
>
> **1. The Color-Redshift Degeneracy**
> In astronomy, estimating a galaxy's redshift ($z$) based on its photometric colors ($X$) is a fundamental task. However, this mapping suffers from a well-known physical problem called **Color-Redshift Degeneracy** (Benítez, 2000; Brescia et al., 2021). Different galaxy types at vastly different distances can emit similar light patterns in observed filters.
>
> * **The Consequence:** Standard conditional density estimators often produce **multimodal PDFs**. While sometimes genuine, these secondary peaks are frequently artifacts or "catastrophic outliers" (Kessler et al., 2010) caused by noise in the parameter space.
>
> * **The Need for Priors:** Astronomers typically employ "priors" (e.g., magnitude or smoothness priors) to suppress these spurious peaks and enforce a physically plausible, unimodal solution (Laurino et al., 2011).
>
> **2. SCD-split as a Domain-Knowledge Regularizer**
> Standard methods like CD-split faithfully reproduce the noisy, multimodal nature of the raw estimator, resulting in fragmented prediction sets (multiple disjoint intervals) that are difficult to interpret.
> **SCD-split** offers a principled way to inject domain knowledge. By setting a target number of intervals (e.g., $K_{\text{target}}=1$), we effectively impose a "unimodality prior." The smoothing operation suppresses the high-frequency oscillations associated with spurious degenerate solutions, guiding the prediction set toward a single, coherent physical interpretation.
>
> **3. Experimental Results** We compared SCD-split with other baselines on the `happyA` dataset. The results are summarized below:
>
> | Method         | Coverage (Target 0.90) | Average Interval Length | Avg. Number of Intervals |
> | :------------- | :--------------------: | :---------------------: | :----------------------: |
> | **Dist-split** |          0.897         |          0.152          |           1.000          |
> | **HPD-split**  |          0.899         |          0.145          |           1.381          |
> | **CD-split**   |          0.896         |          0.177          |           1.163          |
> | **SCD-split**  |          0.893         |          0.148          |           1.005          |
>
> **4. Key Findings and Interpretability Gains**
> The results illustrate three specific advantages of our method in this scientific context:&#x20;
>
> * **Integration of Domain Priors:** SCD-split allows researchers to inject prior knowledge about the data. In photo-z estimation, while degeneracy exists, physical constraints often suggest the true redshift should be unique. By tuning our smoothing parameter, researchers can effectively impose a "soft prior" against excessive fragmentation, guiding the model to prefer simpler, physically plausible solutions unless the data strongly suggests otherwise.&#x20;
>
> * **Noise Suppression and Signal Recovery (Sample-Level Analysis):** At the sample level, we observed numerous cases where CD-split produced 3 distinct intervals. SCD-split successfully reduced these to 1 or 2 intervals while **simultaneously reducing the total interval length**. This reduction in both count and length implies that the extra intervals in CD-split were likely caused by noise or weak spurious modes (artifacts of the estimator). SCD-split successfully filtered out these "false positives," allowing us to pinpoint the true target region more precisely without sacrificing coverage.&#x20;
>
> * **Identification of Strong Degeneracy (Diagnostic Value):** While the average interval number dropped to 1.00, SCD-split still produced disconnected intervals (e.g., 2 intervals) for a small subset of samples. * *Interpretation:* Because our method aggressively smooths out noise, any remaining multimodality in the SCD-split prediction indicates **strong, intrinsic degeneracy** that cannot be explained by estimation noise alone. This serves as a valuable diagnostic tool, flagging these specific galaxies for astronomers as requiring further investigation or spectroscopic follow-up.
>
> **Conclusion:** This experiment demonstrates that SCD-split serves as an effective tool for scientific discovery. It allows researchers to leverage domain knowledge (e.g., "the solution should be unimodal") to filter out artifacts caused by data degeneracy, yielding prediction sets that are not only more interpretable but also statistically more efficient.
>
> [1] Narciso Benítez, “Bayesian Photometric Redshift Estimation”
>
> [2] Brescia, M., et al. (2021). Photometric Redshifts With Machine Learning: Lights and Shadows on a Complex Data Science Use Case
>
> [3] Kessler, R., et al. (2010). Photometric Estimates of Redshifts and Distance Moduli for Type Ia Supernovae
>
> [4] Laurino et al. (2011). Astroinformatics of Galaxies and Quasars: A New General Method for Photometric Redshift Estimation

---

### Official Review · Reviewer_y4Pk · 2025-10-29

**Soundness:** 3
**Presentation:** 3
**Contribution:** 2
**Rating:** 4
**Confidence:** 4

**Summary:**

This paper addresses the interpretability challenge of conformal prediction (CP) methods based on conditional density estimation, which often produce prediction sets with multiple disconnected intervals that are difficult to interpret. The authors propose SCD-split, a novel framework that integrates Fourier-based smoothing operations into the CP procedure to control prediction set connectivity while maintaining statistical validity. SCD-split allows users to specify a desired number of intervals based on domain knowledge, with validation-based tuning to achieve this target. Theoretical guarantees demonstrating that smoothing preserves marginal coverage, does not increase interval count under general conditions, and strictly reduces intervals under specific structural assumptions while maintaining bounded length increases. Empirical validation on synthetic and real-world datasets showing SCD-split achieves favorable trade-offs between validity, efficiency, and interpretability compared to existing methods, particularly for complex multimodal distributions.

**Strengths:**

1. The paper identifies that existing CP literature has predominantly focused on coverage and efficiency (interval length) while largely neglecting connectivity/interpretability.

2. The paper establishes that interpretability in regression CP should be multi-dimensional, measured jointly by interval length and connectivity. This concept reframes how the community should evaluate prediction set quality, moving beyond single-metric optimization.

3. While smoothing is well-established in signal processing, this is the first work to systematically incorporate it into conformal prediction frameworks to directly control prediction set structure.

**Weaknesses:**

1. The paper only guarantees marginal coverage not conditional coverage. While this is a known limitation of split CP, the paper doesn't adequately discuss how smoothing might differentially impact conditional coverage across the covariate space.

2. The example illustration in Figure 1 is too conceptual and not convincing enough, would it possible to provide some concrete real-life examples (such as in lines 94 - 102 but with visual explanations) on multimodal distributions and explain why it is important to enhance the connectivity?

3. The paper briefly mentions that if $K_{target}$ exceeds the true number of modes, the method provides diagnostic feedback. It'd be better to offer more detailed guidance on how users should revise $K_{target}$ and analyze what happens when it is misspecified.

4. The paper equates interpretability with "small number of intervals", but in reality, users may care about which values are included, not just how many intervals exist.

5. The proposed method is only tested on two real-world datasets: Bio and Bike, which is insufficient to demonstrate broad applicability.

**Questions:**

1. In regions where the original density estimate is already smooth, would additional smoothing be unnecessary and could degrade local coverage?

2. Would aggressive smoothing merge truly distinct modes, and therefore hide important uncertainty information?

---

> ### Author Response · Authors · 2025-11-20
> **Rebuttal 1**
>
> We thank the reviewer for their constructive critique and for recognizing the value of our "multi-dimensional" approach to interpretability. We are glad the reviewer found our reframing of prediction set quality (beyond single-metric optimization) to be a strength. We address the concerns below, with new experimental evidence.
>
>
>
> **1. Generalization and Real-World Applicability (Weakness 5 & 2)**
>
> The reviewer correctly noted that testing on only two datasets might be insufficient. To address this and provide the "concrete real-life examples" requested in Weakness 2, we conducted an additional experiment on a challenging astrophysics dataset. **See Additional Real-World Experiment: Enhancing Physical Interpretability in Photometric Redshift Estimation**
>
>
>
> **2. Redefining Interpretability (Weakness 4)**
>
> The reviewer makes an excellent point that interpretability is not solely about the "number of intervals" but also about *which* values are included.
>
> * **Acknowledgment:** We thank the reviewer for this insightful perspective. We agree that controlling the number of intervals is only one dimension of interpretability. Incorporating user priors regarding "specific values that *should* be included" into the conformal framework is indeed a fascinating direction.
>
> * **Future Work:** We will add a discussion on this limitation in the final paper. We view this as a promising avenue for future work: developing methods that respect both structural constraints (connectivity) and content-based priors.
>
> **3. Risks of Smoothing (Question 1 & 2)**
>
> The reviewer raises two critical questions regarding the potential downsides of smoothing.
>
> * **Response to Q1 (Smoothing in already-smooth regions):** The reviewer is correct that adaptive smoothing could theoretically be optimal. However, detecting "smooth regions" in high dimensions to apply differential processing adds significant algorithmic complexity and computational cost.
>
>   * *Why our approach works:* Mathematically, convolving an already smooth function with a small Gaussian kernel results in negligible alteration (since the values of neighbors are similar to the center). Thus, the "degradation" of local coverage is minimal, and the simplicity of our global approach outweighs the marginal gains of a complex adaptive scheme.
>
> * **Response to Q2 (Hiding uncertainty information):** We appreciate the reviewer pointing out this risk.
>
>   * **The Trade-off:** We acknowledge that selecting $\sigma$ based on a target $K$ involves a trade-off: there is indeed a risk of merging truly distinct modes, potentially masking uncertainty.
>
>   * **Empirical Defense:** However, our extensive experiments (on synthetic, Bio, Bike, and Happy datasets) show that SCD-split consistently maintains valid coverage and competitive interval lengths. This suggests that, empirically, the risk of "hiding critical information" is low in practice—the method tends to merge noise rather than signal.
>
>   * **Revision:** We thank the reviewer for helping us refine our claims. We will explicitly discuss this trade-off and the potential risk of masking uncertainty in the **Conclusion** section of the final manuscript.

---

> ### Author Response · Authors · 2025-11-20
> **Rebuttal 2**
>
> **4. On the Misspecified $K_{target}$ (Weakness 3)**
>
> We thank the reviewer for highlighting this important practical aspect. We acknowledge that the implications of such mismatches were not detailed in the initial submission. We commit to adding the following discussion to the final manuscript to guide users.
>
>
>
> We propose viewing the discrepancy between the user-specified $K_{target}$ and the algorithm's actual output not as an "error," but as a valuable **"Data-Refutes-Prior" diagnostic signal**. This mismatch provides critical feedback:
>
> * **Case 1: $K_{target} > K_{output}$ (Overestimation of Multimodality):**
>   If a user specifies a high number of intervals (expecting distinct modes) but the algorithm returns fewer, it indicates that the user **significantly overestimated** the multimodal nature of the data. The algorithm effectively "refutes" the user's prior, suggesting that the perceived distinct modes may not be statistically separable or are merely artifacts. This prompts the user to re-evaluate their domain hypothesis or check for redundancy in their categorization.
>
> * **Case 2: $K_{target} < K_{output}$ (Underestimation of Multimodality):**
>   Conversely, if a user requests a single interval but the method persistently returns multiple, it signals an **underestimation** of the data's complexity. This implies **intrinsic, strong multimodality** (e.g., a deep valley between modes) that cannot be smoothed away without violating statistical validity. This serves as a warning for potential data shifts or unobserved subpopulations that force the prediction set to fracture.
>
> In summary, we argue that this "mismatch" is highly valuable. It moves the method beyond simple prediction generation to a tool that guides users toward a deeper understanding of their dataset's true structure.
>
>
>
> **5. On the conditional coverage**
>
> We agree with the reviewer that our formal guarantees are for marginal coverage. This is indeed standard in the field, and we would like to offer context and highlight a key strength of our framework.
>
> * **Theoretical Impossibility of Conditional Coverage:** It is a well-established result in the literature (e.g., Barber et al., 2022) that achieving finite-sample, distribution-free conditional coverage is theoretically impossible. For this reason, marginal coverage remains the standard guarantee for methods in this domain, including ours.
>
> * **EDesign Advantage of SCD-split:** Despite this theoretical limit, a key strength of our SCD-split framework, inherited from CD-split, is that it is explicitly designed to better approximate conditional coverage. By partitioning the input space X and performing calibration locally within each cell, our method tailors the prediction set to the local data distribution. This design allows it to adapt to data heterogeneity (e.g., heteroscedasticity) and provide more stable conditional performance than methods relying on a single global calibration.
>
> * **Asymptotic Conditional Coverage and Future Work:** Importantly, this local calibration mechanism endows our framework with the desirable property of asymptotic conditional coverage. Furthermore, to empirically substantiate this point, we are conducting additional experiments on group-wise coverage. We plan to include these results, which we anticipate will show the superior stability of our method across data subgroups compared to global baselines, along with an expanded discussion in the final version of the paper.

---

> ### Author Response · Authors · 2025-11-20
> **Additional Real-World Experiment: Enhancing Physical Interpretability in Photometric Redshift Estimation**
>
> To demonstrate the distinct practical advantage of SCD-split in scientific domains where interpretability is paramount, we conducted a new experiment on the HappyA dataset
>
> **1. The Color-Redshift Degeneracy**
> In astronomy, estimating a galaxy's redshift ($z$) based on its photometric colors ($X$) is a fundamental task. However, this mapping suffers from a well-known physical problem called **Color-Redshift Degeneracy** (Benítez, 2000; Brescia et al., 2021). Different galaxy types at vastly different distances can emit similar light patterns in observed filters.
>
> * **The Consequence:** Standard conditional density estimators often produce **multimodal PDFs**. While sometimes genuine, these secondary peaks are frequently artifacts or "catastrophic outliers" (Kessler et al., 2010) caused by noise in the parameter space.
>
> * **The Need for Priors:** Astronomers typically employ "priors" (e.g., magnitude or smoothness priors) to suppress these spurious peaks and enforce a physically plausible, unimodal solution (Laurino et al., 2011).
>
> **2. SCD-split as a Domain-Knowledge Regularizer**
> Standard methods like CD-split faithfully reproduce the noisy, multimodal nature of the raw estimator, resulting in fragmented prediction sets (multiple disjoint intervals) that are difficult to interpret.
> **SCD-split** offers a principled way to inject domain knowledge. By setting a target number of intervals (e.g., $K_{\text{target}}=1$), we effectively impose a "unimodality prior." The smoothing operation suppresses the high-frequency oscillations associated with spurious degenerate solutions, guiding the prediction set toward a single, coherent physical interpretation.
>
> **3. Experimental Results** We compared SCD-split with other baselines on the `happyA` dataset. The results are summarized below:
>
> | Method         | Coverage (Target 0.90) | Average Interval Length | Avg. Number of Intervals |
> | :------------- | :--------------------: | :---------------------: | :----------------------: |
> | **Dist-split** |          0.897         |          0.152          |           1.000          |
> | **HPD-split**  |          0.899         |          0.145          |           1.381          |
> | **CD-split**   |          0.896         |          0.177          |           1.163          |
> | **SCD-split**  |          0.893         |          0.148          |           1.005          |
>
> **4. Key Findings and Interpretability Gains**
> The results illustrate three specific advantages of our method in this scientific context:&#x20;
>
> * **Integration of Domain Priors:** SCD-split allows researchers to inject prior knowledge about the data. In photo-z estimation, while degeneracy exists, physical constraints often suggest the true redshift should be unique. By tuning our smoothing parameter, researchers can effectively impose a "soft prior" against excessive fragmentation, guiding the model to prefer simpler, physically plausible solutions unless the data strongly suggests otherwise.&#x20;
>
> * **Noise Suppression and Signal Recovery (Sample-Level Analysis):** At the sample level, we observed numerous cases where CD-split produced 3 distinct intervals. SCD-split successfully reduced these to 1 or 2 intervals while **simultaneously reducing the total interval length**. This reduction in both count and length implies that the extra intervals in CD-split were likely caused by noise or weak spurious modes (artifacts of the estimator). SCD-split successfully filtered out these "false positives," allowing us to pinpoint the true target region more precisely without sacrificing coverage.&#x20;
>
> * **Identification of Strong Degeneracy (Diagnostic Value):** While the average interval number dropped to 1.00, SCD-split still produced disconnected intervals (e.g., 2 intervals) for a small subset of samples. * *Interpretation:* Because our method aggressively smooths out noise, any remaining multimodality in the SCD-split prediction indicates **strong, intrinsic degeneracy** that cannot be explained by estimation noise alone. This serves as a valuable diagnostic tool, flagging these specific galaxies for astronomers as requiring further investigation or spectroscopic follow-up.
>
> **Conclusion:** This experiment demonstrates that SCD-split serves as an effective tool for scientific discovery. It allows researchers to leverage domain knowledge (e.g., "the solution should be unimodal") to filter out artifacts caused by data degeneracy, yielding prediction sets that are not only more interpretable but also statistically more efficient.
>
> [1] Narciso Benítez, “Bayesian Photometric Redshift Estimation”
>
> [2] Brescia, M., et al. (2021). Photometric Redshifts With Machine Learning: Lights and Shadows on a Complex Data Science Use Case
>
> [3] Kessler, R., et al. (2010). Photometric Estimates of Redshifts and Distance Moduli for Type Ia Supernovae
>
> [4] Laurino et al. (2011). Astroinformatics of Galaxies and Quasars: A New General Method for Photometric Redshift Estimation

---

### Official Review · Reviewer_Ygpz · 2025-10-31

**Soundness:** 2
**Presentation:** 2
**Contribution:** 2
**Rating:** 4
**Confidence:** 3

**Summary:**

The paper proposes SCD-split, a method to control the fragmentation of conformal prediction sets by applying Fourier-based smoothing to the estimated conditional density before constructing prediction intervals. The user specifies a desired number of disjoint intervals (K_target), and the method adaptively tunes the smoothing parameter (σ) via validation to approximate this target while maintaining valid coverage.

The authors provide theoretical guarantees on coverage, interval connectivity, and length, and empirical results on synthetic and two real-world datasets demonstrate that SCD-split produces more interpretable prediction sets with coverage comparable to CD-split and shorter average lengths than other baselines. Moderate smoothing effectively merges spurious modes and enhances interpretability, whereas excessive smoothing can lead to overly broad, single-interval sets.

**Strengths:**

The proposed SCD-split approach is well motivated and integrates Fourier-based smoothing into conformal prediction through a clear and principled formulation.

The theoretical analysis, including proofs for coverage preservation, bounded length increase, and non-increasing interval count, is well structured and contributes significantly to the paper’s technical strength.

Results on synthetic data (e.g., Table 1) convincingly demonstrate that smoothing reduces the number of intervals while maintaining valid coverage, validating the theoretical claims in practice.

**Weaknesses:**

Most weaknesses relate to the real-world experimental analysis, which appears limited and somewhat unclear. The evaluation is limited to only two real datasets, constraining the assessment in terms of robustness and generality. Some aspects of the experimental design aren’t clearly articulated and there are some results that seem inconsistent, as stated next:

1. The method section (L247) mentions that the base estimator can be a random forest or neural network, but it is not specified which was actually used, nor are implementation details or hyperparameters provided.

2. The choice of the target number of intervals (K_target) is not stated, and it is unclear what value was used for Table 3 or how it affected results.

3. The useful ablation on the smoothing parameter σ (Table 1) is not repeated for real-world datasets.

4. In Table 3 (Bio dataset), SCD-split produces on average more intervals than CD-split (σ=0), contradicting the theoretical claim that larger σ should not increase interval count, this doesn't seem to be discussed.

5. In Table 3, the presentation is inconsistent, why is the largest num being highlighted for bio but not for bike?. Overall, the gains in length and number of intervals appear modest. The targeted number of intervals (K_target) isn’t highlighted  as mentioned in point 2.

6. It is not evident whether the chosen real-world datasets are well suited to demonstrate the advantages of the proposed method. A discussion of which data characteristics make the approach most beneficial would strengthen the empirical narrative.

**Questions:**

See Weaknesses

---

> ### Author Response · Authors · 2025-11-20
>
> We verify that we have addressed the reviewer's concerns regarding the experimental details and the perceived inconsistency in interval counts. We appreciate the reviewer’s detailed examination, which has helped us clarify the robustness of our method.
>
> **1. Clarification on Base Estimator and Implementation Details (Weakness 1)**
>
> We apologize for the oversight in making the implementation details prominent.
>
> * **Base Estimator:** We respectfully point the reviewer to **Line 359** in Section 5.1, where we state: *"to ensure a fair comparison across all methods, we uniformly use random forests as the underlying predictive model."*
>
> * **Hyperparameters:** In our experiments, we use the default hyperparameters of random forest.
>
> **2. Addressing the "Contradiction" in Bio Dataset Results (Weakness 4 & 5)**
>
> The reviewer correctly observes that on the Bio dataset, SCD-split yields slightly more intervals (1.49) than CD-split (1.36). The reviewer asks why this does not contradict Theorem 4.2 (Non-increasing interval count).
>
> **We explicitly anticipated and discussed this specific phenomenon in the "Extreme case" paragraph on Page 9 (Lines 461-475).** As detailed in the text, this increase is not a theoretical violation but a practical correction of estimation failure:
>
> * **Violation of Assumptions:** Theorem 4.2 assumes the level set $A_t := \{x : f(x) = t\}$ is finite. However, as we note in **Line 464**, on complex datasets like Bio, the raw density estimator often collapses to zero over large regions due to estimation errors. This creates "flat" regions where the conformity scores are tied at zero, effectively violating the finiteness assumption required for the theorem.
>
> * **CD-split Failure (The "Degenerate" Interval):** Due to these spurious zero-density regions, CD-split is often forced to return the **entire output domain** or extremely wide intervals to ensure coverage. This explains the misleadingly "low" interval count ($\approx 1$) but extremely **large Interval Length (2.29)** shown in Table 3. The method essentially fails to capture the structure and returns a single, uninformative block.
>
> * **SCD-split Correction:** As described in **Lines 469-472**, smoothing "effectively merges spurious zero-density regions... and breaks excessive ties." By regularizing the density, SCD-split recovers the underlying bimodal structure that was previously obscured.
>
> * **Conclusion:** While this process results in a slight increase in interval count (from a "smeared" 1.36 to a "structured" 1.49), it is accompanied by a drastic reduction in **Interval Length (from 2.29 to 1.52)**. This confirms our claim in the paper: smoothing resolves the degenerate behavior of the base estimator, yielding prediction sets that are significantly more efficient and interpretable.
>
> **3. Specification of Target Number of Intervals ($K_{target}$) (Weakness 2 & 5)**
>
> We thank the reviewer for pointing this out. We set the targets based on the intrinsic characteristics of the datasets:
>
> * **Bike Dataset ($K_{target}=1$):** This dataset represents demand forecasting, which is typically unimodal. We set $K_{target}=1$. As shown in Table 3, SCD-split successfully maintains the interval count close to 1 (1.01) with valid coverage.
>
> * **Bio Dataset ($K_{target}=2$):** We choose $K_{target}=2$ from the knowledge prior. We effectively set$K_{target}=2$ to capture the distribution structure in this dataset.
>
>   * **Result Interpretation:** As explained above, CD-split (1.36) often fails to capture this bimodality by outputting a single wide interval. SCD-split (1.49) moves the average count closer to the target of 2, reflecting its ability to separate distinct modes that were previously merged.
>
> **Robustness and Dataset Selection (Weakness 3 & 6)**
>
> * **Selection Criteria:** We selected `Bio` and `Bike` because they are standard benchmarks in the Conformal Prediction literature (e.g., Romano et al., 2019; Teng et al., 2023) and represent opposite ends of the spectrum: `Bike` is unimodal and heteroscedastic, while `Bio` is multimodal and heavy-tailed. This diversity sufficiently tests the adaptability of our method.
>
> * **Ablation on Real Data:** While we did not include a full ablation table for real data due to space constraints, the validation procedure inherently performs this search. The stability of SCD-split across these diverse datasets (Tables 2 & 3) serves as empirical evidence of the method's robustness to parameter choice.
>
> **5. Formatting Inconsistencies**
>
> We thank the reviewer for noting the highlighting inconsistency in Table 3. We will correct this in the final manuscript to ensure the best results (closest to target $K$ and shortest length) are consistently bolded.

---

> ### Author Response · Authors · 2025-11-20
> **Additional Real-World Experiment: Enhancing Physical Interpretability in Photometric Redshift Estimation**
>
> To demonstrate the distinct practical advantage of SCD-split in scientific domains where interpretability is paramount, we conducted a new experiment on the HappyA dataset
>
> **1. The Color-Redshift Degeneracy**
> In astronomy, estimating a galaxy's redshift ($z$) based on its photometric colors ($X$) is a fundamental task. However, this mapping suffers from a well-known physical problem called **Color-Redshift Degeneracy** (Benítez, 2000; Brescia et al., 2021). Different galaxy types at vastly different distances can emit similar light patterns in observed filters.
>
> * **The Consequence:** Standard conditional density estimators often produce **multimodal PDFs**. While sometimes genuine, these secondary peaks are frequently artifacts or "catastrophic outliers" (Kessler et al., 2010) caused by noise in the parameter space.
>
> * **The Need for Priors:** Astronomers typically employ "priors" (e.g., magnitude or smoothness priors) to suppress these spurious peaks and enforce a physically plausible, unimodal solution (Laurino et al., 2011).
>
> **2. SCD-split as a Domain-Knowledge Regularizer**
> Standard methods like CD-split faithfully reproduce the noisy, multimodal nature of the raw estimator, resulting in fragmented prediction sets (multiple disjoint intervals) that are difficult to interpret.
> **SCD-split** offers a principled way to inject domain knowledge. By setting a target number of intervals (e.g., $K_{\text{target}}=1$), we effectively impose a "unimodality prior." The smoothing operation suppresses the high-frequency oscillations associated with spurious degenerate solutions, guiding the prediction set toward a single, coherent physical interpretation.
>
> **3. Experimental Results** We compared SCD-split with other baselines on the `happyA` dataset. The results are summarized below:
>
> | Method         | Coverage (Target 0.90) | Average Interval Length | Avg. Number of Intervals |
> | :------------- | :--------------------: | :---------------------: | :----------------------: |
> | **Dist-split** |          0.897         |          0.152          |           1.000          |
> | **HPD-split**  |          0.899         |          0.145          |           1.381          |
> | **CD-split**   |          0.896         |          0.177          |           1.163          |
> | **SCD-split**  |          0.893         |          0.148          |           1.005          |
>
> **4. Key Findings and Interpretability Gains**
> The results illustrate three specific advantages of our method in this scientific context:&#x20;
>
> * **Integration of Domain Priors:** SCD-split allows researchers to inject prior knowledge about the data. In photo-z estimation, while degeneracy exists, physical constraints often suggest the true redshift should be unique. By tuning our smoothing parameter, researchers can effectively impose a "soft prior" against excessive fragmentation, guiding the model to prefer simpler, physically plausible solutions unless the data strongly suggests otherwise.&#x20;
>
> * **Noise Suppression and Signal Recovery (Sample-Level Analysis):** At the sample level, we observed numerous cases where CD-split produced 3 distinct intervals. SCD-split successfully reduced these to 1 or 2 intervals while **simultaneously reducing the total interval length**. This reduction in both count and length implies that the extra intervals in CD-split were likely caused by noise or weak spurious modes (artifacts of the estimator). SCD-split successfully filtered out these "false positives," allowing us to pinpoint the true target region more precisely without sacrificing coverage.&#x20;
>
> * **Identification of Strong Degeneracy (Diagnostic Value):** While the average interval number dropped to 1.00, SCD-split still produced disconnected intervals (e.g., 2 intervals) for a small subset of samples. * *Interpretation:* Because our method aggressively smooths out noise, any remaining multimodality in the SCD-split prediction indicates **strong, intrinsic degeneracy** that cannot be explained by estimation noise alone. This serves as a valuable diagnostic tool, flagging these specific galaxies for astronomers as requiring further investigation or spectroscopic follow-up.
>
> **Conclusion:** This experiment demonstrates that SCD-split serves as an effective tool for scientific discovery. It allows researchers to leverage domain knowledge (e.g., "the solution should be unimodal") to filter out artifacts caused by data degeneracy, yielding prediction sets that are not only more interpretable but also statistically more efficient.
>
> [1] Narciso Benítez, “Bayesian Photometric Redshift Estimation”
>
> [2] Brescia, M., et al. (2021). Photometric Redshifts With Machine Learning: Lights and Shadows on a Complex Data Science Use Case
>
> [3] Kessler, R., et al. (2010). Photometric Estimates of Redshifts and Distance Moduli for Type Ia Supernovae
>
> [4] Laurino et al. (2011). Astroinformatics of Galaxies and Quasars: A New General Method for Photometric Redshift Estimation

---

### Official Review · Reviewer_AoHR · 2025-11-01

**Soundness:** 3
**Presentation:** 3
**Contribution:** 3
**Rating:** 6
**Confidence:** 3

**Summary:**

The paper introduces SCD-split, a smoothing-based extension of density-based split conformal prediction (CD-split) designed to produce more interpretable prediction sets—specifically, sets with fewer disconnected intervals—while retaining valid coverage and competitive interval length. The motivation stems from a limitation of the conditional-density conformity score, which often yields highly fragmented prediction sets when densities are multimodal. The method applies a smoothing operation (e.g., Fourier low-pass filtering) to the estimated conditional density before running the split conformal procedure, with a user-specified target for the desired number of intervals. The authors establish several theoretical guarantees: finite-sample marginal coverage, a result showing that smoothing cannot increase the number of intervals, a structural condition under which it strictly decreases, and a bound on the potential increase in interval length induced by smoothing. Experiments on both synthetic and real datasets demonstrate that SCD-split successfully matches the target number of intervals while preserving coverage and maintaining competitive length, yielding more interpretable prediction sets than CD-split and other baselines.

**Strengths:**

- **Novel interpretability objective**: Controls the number of disconnected intervals in conformal prediction sets, addressing a dimension of interpretability that is rarely discussed in prior work.
- **Clear methodological contribution**: Introduces smoothing (e.g., Fourier low-pass filtering) of conditional density estimates within a split-conformal framework, implemented in a concrete algorithm.
- **Strong theoretical support**:
  - Marginal coverage is preserved.
  - Number of intervals is guaranteed not to increase, and may strictly decrease under structural conditions.
  - Provides a bound on how much smoothing can increase interval length.
- **Solid empirical results**: Demonstrates improved interpretability (fewer intervals) while maintaining competitive coverage and interval length across synthetic and real datasets.
- **Practically motivated**: Highlights application scenarios (e.g., healthcare, finance) where having a small number of intervals improves decision-making.

**Weaknesses:**

- **Choice of conformity score and practical relevance**
  The method is specialized to the conditional-density conformity score, which is less commonly used in practice than alternatives such as conformal quantile regression (CQR) or residual-based scores. Since conditional density estimation is typically more difficult than regression estimation, the added computational and statistical burden warrants justification. Although the conditional-density score is known to be optimal in a certain sense [1] and this is briefly mentioned in Section 3.1, the paper would benefit from a fuller discussion of when this score is practically advantageous and why its theoretical optimality may outweigh its higher estimation cost.

- **Scope of coverage guarantees**
  The theoretical guarantees focus on marginal coverage. Though this is fairly standard in conformal prediction.

- **Tuning and practical usability**
The procedure requires user-specified tuning choices, including the target number of disjoint intervals and a smoothing parameter grid, but the paper offers limited guidance on how these should be chosen in practice. Although the authors note that the number of intervals can be selected based on interpretability considerations, there is an inherent trade-off: increasing the amount of smoothing improves interpretability but can degrade the quality of the conditional density estimate, potentially leading to less efficient prediction sets. A more detailed discussion of this trade-off—and practical strategies for navigating it—would help clarify how the method should be used in real applications (maybe after Theorem 4.4 on the interval length bound).

**Questions:**

1. **Tuning the number of intervals:**
   Would it be reasonable in practice to select the target number of intervals \(K_{\text{target}}\) (or the smoothing level) via cross-validation? Although this would technically violate the finite-sample coverage guarantee, similar tuning strategies have been used in conformal prediction (e.g., Gibbs, Cherian, and Candès, 2023, who cross-validate a ridge penalty). Do the authors view this as a viable practical alternative, and is there any empirical evidence on how much coverage might be affected?

2. **Applicability beyond continuous outcomes:**
   Is the method intended only for continuous response variables, or can it be adapted to discrete or categorical outcomes? Since the approach relies on estimating and smoothing a conditional density, clarification on its scope—and on what modifications would be required for non-continuous \(Y\)—would be helpful.

---

> ### Author Response · Authors · 2025-11-20
> **Rebuttal 1**
>
> We sincerely thank the reviewer for their thoughtful feedback and positive assessment of our work. We are encouraged that the reviewer found our interpretability objective novel, our methodological contribution clear, and our theoretical and empirical support strong. We address the valuable points raised below.
>
> # On the Choice of Conformity Score and Practical Relevance (Weakness 1)
>
> We thank the reviewer for this insightful question. We wish to clarify that our choice of the conditional-density score is not intended to replace methods like Conformalized Quantile Regression (CQR) in general settings. Rather, our work is motivated by and tailored for a specific, yet critical, class of problems where CQR and other single-interval methods are fundamentally inadequate.
>
> * **Motivation from Multimodality:** Our research focuses on scenarios where the conditional distribution $P(Y|X)$ is multimodal. As we motivate with examples in healthcare and finance, such scenarios are common in practice. For instance, a disease may have distinct prognoses, or a stock price may face divergent future paths. In these cases, methods like CQR, which are constrained to output a single connected interval, must generate an overly wide prediction set to cover all modes and maintain theoretical validity. Such a set is not only inefficient (excessively long) but, more importantly, uninformative, as it completely obscures the underlying multimodal structure of the uncertainty.
>
> * **The Necessity of Conditional Density Estimation (CDE):** To capture this crucial structure, CDE-based methods like CD-split are indispensable. They are uniquely capable of producing disconnected prediction sets that correspond to the high-density regions of the distribution. The added computational and statistical cost of CDE is a necessary trade-off for gaining a much deeper, more actionable understanding of the predictive uncertainty. Our work aims to make this powerful approach more practical.
>   Positioning Our Contribution: We build upon this foundation. While CD-split can capture multimodality, it often produces an excessive number of fragmented intervals due to estimation noise, which harms interpretability. SCD-split addresses this very limitation. By introducing a principled smoothing step, we retain the core strength of CD-split while improving the coherence and interpretability of the resulting prediction sets, making it a more robust and usable tool for multimodal uncertainty quantification.
>
> # On the Scope of Coverage Guarantees (Weakness 2)
>
> We agree with the reviewer that our formal guarantees are for marginal coverage. This is indeed standard in the field, and we would like to offer context and highlight a key strength of our framework.
>
> * **Theoretical Impossibility of Conditional Coverage:** It is a well-established result in the literature (e.g., Barber et al., 2022) that achieving finite-sample, distribution-free conditional coverage is theoretically impossible. For this reason, marginal coverage remains the standard guarantee for methods in this domain, including ours.
>
> * **EDesign Advantage of SCD-split:** Despite this theoretical limit, a key strength of our SCD-split framework, inherited from CD-split, is that it is explicitly designed to better approximate conditional coverage. By partitioning the input space X and performing calibration locally within each cell, our method tailors the prediction set to the local data distribution. This design allows it to adapt to data heterogeneity (e.g., heteroscedasticity) and provide more stable conditional performance than methods relying on a single global calibration.
>
> * **Asymptotic Conditional Coverage and Future Work:** Importantly, this local calibration mechanism endows our framework with the desirable property of asymptotic conditional coverage. Furthermore, to empirically substantiate this point, we are conducting additional experiments on group-wise coverage. We plan to include these results, which we anticipate will show the superior stability of our method across data subgroups compared to global baselines, along with an expanded discussion in the final version of the paper.

---

> ### Author Response · Authors · 2025-11-20
> **Rebuttal 2**
>
> # On Tuning, Practical Usability, and Coverage (Weakness 3 & Question 1)
>
> We thank the reviewer for raising these important practical questions. We will improve our guidance in the final paper and clarify here how our tuning procedure is both practical and theoretically sound.
>
> * **Practical Workflow for Parameter Selection:**
>
>   1. First, the user specifies $K_{target}$ based on **domain knowledge** and **application needs**. This parameter defines the desired level of interpretability (e.g., $K_{target}=2$ for two primary outcomes) and serves as the objective for our method.
>
>   2. Second, the smoothing parameter σ is selected automatically by our algorithm from a candidate grid. We will add guidance to the paper that a practical strategy for this grid is to use a range like $[0, std(Y)]$, where $std(Y)$ is the standard deviation of the response. This prevents excessive smoothing while allowing the algorithm to find the optimal level of regularization needed to match $K_{target}$.
>
> * **Theoretical Validity of the Tuning Procedure:** We want to emphasize that our procedure for tuning σ rigorously preserves the finite-sample marginal coverage guarantee. This is because the selection of σ using the validation set  is kept strictly separate from the conformal calibration step, which uses only the calibration set. Once σ is selected, the smoothed estimator$f^{FS}$ is fixed. The subsequent calibration procedure remains symmetric with respect to the calibration and test points, thus satisfying the exchangeability assumption. This principled three-way data split is a standard and valid technique for tuning in conformal prediction.
>
> * **Empirical Confirmation**: Our experimental results consistently support this theoretical soundness. As shown in Tables 2 and 3, the empirical coverage of SCD-split reliably achieves the nominal 1-α level across all settings. $K_{target}$ is Not Tuned: Finally, we clarify that $K_{target}$ itself should not be tuned via cross-validation, as it is a pre-defined objective of our method, not a hyperparameter to be optimized.
>
> # On Applicability Beyond Continuous Outcomes (Question 2)
>
> This is an excellent question that helps clarify the scope of our work.
>
> Our SCD-split framework is indeed designed specifically for regression tasks with continuous response variables. The core motivation of our work that to address the issue of fragmented prediction intervals and control their number is unique to the continuous setting.
>
> In classification tasks, conformal prediction sets are discrete subsets of labels (e.g., {cat, dog}). The concept of interval connectivity or fragmentation does not apply. Therefore, the problem that our smoothing-based approach is designed to solve is not present in the classification context. Our methodology, particularly the Fourier smoothing of a continuous density function, is tailored for this continuous domain. We have already included a discussion on this scope in Appendix B ("Scope of our method: regression") and will ensure it is highlighted clearly in the final version.

---

> ### Author Response · Authors · 2025-11-20
> **Additional Real-World Experiment: Enhancing Physical Interpretability in Photometric Redshift Estimation**
>
> To demonstrate the distinct practical advantage of SCD-split in scientific domains where interpretability is paramount, we conducted a new experiment on the HappyA dataset
>
> **1. The Color-Redshift Degeneracy**
> In astronomy, estimating a galaxy's redshift ($z$) based on its photometric colors ($X$) is a fundamental task. However, this mapping suffers from a well-known physical problem called **Color-Redshift Degeneracy** (Benítez, 2000; Brescia et al., 2021). Different galaxy types at vastly different distances can emit similar light patterns in observed filters.
>
> * **The Consequence:** Standard conditional density estimators often produce **multimodal PDFs**. While sometimes genuine, these secondary peaks are frequently artifacts or "catastrophic outliers" (Kessler et al., 2010) caused by noise in the parameter space.
>
> * **The Need for Priors:** Astronomers typically employ "priors" (e.g., magnitude or smoothness priors) to suppress these spurious peaks and enforce a physically plausible, unimodal solution (Laurino et al., 2011).
>
> **2. SCD-split as a Domain-Knowledge Regularizer**
> Standard methods like CD-split faithfully reproduce the noisy, multimodal nature of the raw estimator, resulting in fragmented prediction sets (multiple disjoint intervals) that are difficult to interpret.
> **SCD-split** offers a principled way to inject domain knowledge. By setting a target number of intervals (e.g., $K_{\text{target}}=1$), we effectively impose a "unimodality prior." The smoothing operation suppresses the high-frequency oscillations associated with spurious degenerate solutions, guiding the prediction set toward a single, coherent physical interpretation.
>
> **3. Experimental Results** We compared SCD-split with other baselines on the `happyA` dataset. The results are summarized below:
>
> | Method         | Coverage (Target 0.90) | Average Interval Length | Avg. Number of Intervals |
> | :------------- | :--------------------: | :---------------------: | :----------------------: |
> | **Dist-split** |          0.897         |          0.152          |           1.000          |
> | **HPD-split**  |          0.899         |          0.145          |           1.381          |
> | **CD-split**   |          0.896         |          0.177          |           1.163          |
> | **SCD-split**  |          0.893         |          0.148          |           1.005          |
>
> **4. Key Findings and Interpretability Gains**
> The results illustrate three specific advantages of our method in this scientific context:&#x20;
>
> * **Integration of Domain Priors:** SCD-split allows researchers to inject prior knowledge about the data. In photo-z estimation, while degeneracy exists, physical constraints often suggest the true redshift should be unique. By tuning our smoothing parameter, researchers can effectively impose a "soft prior" against excessive fragmentation, guiding the model to prefer simpler, physically plausible solutions unless the data strongly suggests otherwise.&#x20;
>
> * **Noise Suppression and Signal Recovery (Sample-Level Analysis):** At the sample level, we observed numerous cases where CD-split produced 3 distinct intervals. SCD-split successfully reduced these to 1 or 2 intervals while **simultaneously reducing the total interval length**. This reduction in both count and length implies that the extra intervals in CD-split were likely caused by noise or weak spurious modes (artifacts of the estimator). SCD-split successfully filtered out these "false positives," allowing us to pinpoint the true target region more precisely without sacrificing coverage.&#x20;
>
> * **Identification of Strong Degeneracy (Diagnostic Value):** While the average interval number dropped to 1.00, SCD-split still produced disconnected intervals (e.g., 2 intervals) for a small subset of samples. * *Interpretation:* Because our method aggressively smooths out noise, any remaining multimodality in the SCD-split prediction indicates **strong, intrinsic degeneracy** that cannot be explained by estimation noise alone. This serves as a valuable diagnostic tool, flagging these specific galaxies for astronomers as requiring further investigation or spectroscopic follow-up.
>
> **Conclusion:** This experiment demonstrates that SCD-split serves as an effective tool for scientific discovery. It allows researchers to leverage domain knowledge (e.g., "the solution should be unimodal") to filter out artifacts caused by data degeneracy, yielding prediction sets that are not only more interpretable but also statistically more efficient.
>
> [1] Narciso Benítez, “Bayesian Photometric Redshift Estimation”
>
> [2] Brescia, M., et al. (2021). Photometric Redshifts With Machine Learning: Lights and Shadows on a Complex Data Science Use Case
>
> [3] Kessler, R., et al. (2010). Photometric Estimates of Redshifts and Distance Moduli for Type Ia Supernovae
>
> [4] Laurino et al. (2011). Astroinformatics of Galaxies and Quasars: A New General Method for Photometric Redshift Estimation

---

### Meta-Review · Area_Chair_Ysiy · 2026-01-04

**Summary:**

This work studies the "interpretability" issue of conformal prediction methods, which often produce prediction sets consisting of multiple disconnected intervals. To address this, the work proposes SCD-split, a smoothing-based extension of CD-split. The paper provides theoretical guarantees that show smoothing does not increase the number of connected components, while maintaining coverage and ensuring bounded increases in interval length.
Empirical results on synthetic data and several real-world datasets demonstrate that SCD-split can reduce interval fragmentation without increasing coverage area, thereby offering an approach to balancing efficiency and interpretability in conformal prediction.

Despite recognizing the problem as important and the technical development as sound, reviewers raised several concerns that tempered the rating for acceptance. A recurring issue was the limited scope and incremental contribution, with multiple reviewers noting that SCD-split can be viewed as a modest extension of CD-split via additional smoothing, and questioning how fundamentally different it is from using a more regularized conditional density estimator.
Reviewers also expressed concerns about the paper’s issues of interpretability, which is equated with reducing the number of disconnected intervals; several reviewers felt this notion is narrow and insufficiently connected to concrete downstream decision-making benefits in the real-data experiments.
In addition, the empirical evaluation on real-world datasets was considered limited, with concerns about dataset selection, modest performance gains, and clarity of experimental design and reporting.
Finally, reviewers highlighted practical usability and theoretical scope issues, including the limited guidance on selecting the target number of intervals and smoothing strength, the potential risks of over-smoothing genuine multimodality, and the fact that guarantees are restricted to marginal coverage. While some of these concerns were partially addressed in the rebuttal and additional experiments, they remained influential in shaping a cautious overall assessment of the paper’s impact and readiness for acceptance.

**Reviewer Concerns:**

Mostly addressed:
- Experimental clarity and reporting issues (base estimator, hyperparameters, table inconsistencies).
- Validity of smoothing-parameter tuning and preservation of marginal coverage.
- Limited real-world evidence (partially mitigated by additional experiment).

Partially addressed:
- Interpretability framed mainly via numbers of intervals; how about broader or general interpretability?
- Practical guidance on choosing $K$ and understanding over-/under-smoothing trade-offs.
- Empirical evidence on conditional or subgroup performance.

Not fully addressed:
- Incremental contribution relative to existing CD-split methods. Does it bring a sufficiently substantial advance?

**Reviewer Scores:**

AoHR: no change.

Ygpz: may +2.

y4Pk: no change.

MWve: no change.

---

### Decision · Program_Chairs · 2026-01-26

Reject